# CLINICALLAB: ALIGNING AGENTS FOR MULTI-DEPARTMENTAL CLINICAL DIAGNOSTICS IN THE REAL WORLD

## ABSTRACT

Large language models (LLMs) have achieved significant performance progress in various natural language processing applications. However, LLMs still struggle to meet the strict requirements for accuracy and reliability in the medical field and face many challenges in clinical applications. Existing clinical diagnostic evaluation benchmarks for evaluating medical agents powered by LLMs have severe limitations. Firstly, most existing medical evaluation benchmarks face the risk of data leakage or contamination. Secondly, existing benchmarks often neglect the characteristics of multiple departments and specializations in modern medical practice. Thirdly, existing evaluation methods are limited to multiple-choice questions, which do not align with the real-world diagnostic scenarios. Lastly, existing evaluation methods lack comprehensive evaluations of end-to-end real clinical scenarios. These limitations in benchmarks in turn obstruct advancements of LLMs and agents for medicine. To address these limitations, we introduce **ClinicalLab**, a comprehensive clinical diagnosis agent alignment suite. ClinicalLab includes **ClinicalBench**, an end-to-end multi-departmental clinical diagnostic evaluation benchmark for evaluating medical agents and LLMs. ClinicalBench is based on real cases that cover 24 departments and 150 diseases. We ensure that ClinicalBench does not have data leakage. ClinicalLab also includes four novel metrics (**ClinicalMetrics**) for evaluating the effectiveness of LLMs in clinical diagnostic tasks. We evaluate 17 general and medical-domain LLMs and find that their performance varies significantly across different departments. Based on these findings, in ClinicalLab, we propose **ClinicalAgent**, an end-to-end clinical agent that aligns with real-world clinical diagnostic practices. We systematically investigate the performance and applicable scenarios of variants of ClinicalAgent on ClinicalBench. Our findings demonstrate the importance of aligning with modern medical practices in designing medical agents.

## 1 INTRODUCTION

Large Language Models (LLMs) have demonstrated exceptional capabilities across a wide variety of natural language processing tasks (Hendrycks et al., 2021; Cobbe et al., 2021; Song et al., 2024). Recent studies attempt to apply LLMs to the medical field (Singhal et al., 2023), aiming to improve the efficiency of healthcare systems through early disease diagnosis and timely health management, thereby reducing the workload of medical professionals. However, the high sensitivity of the medical field and the strict requirements for accuracy and reliability in clinical decision-making pose great challenges to LLMs. Recent studies find that LLMs are prone to producing hallucinations in medical diagnoses (Singhal et al., 2022; Bubeck et al., 2023), and such erroneous diagnoses could harm the physical and psychological health of patients and also potentially lead to serious medical negligence. How to effectively, reliably, and comprehensively evaluate the true capabilities of LLMs and the accuracy of their diagnostic results, as well as reveal their limitations to avoid potential misdiagnosis, remains an unresolved problem.

Recent studies find that existing benchmarks cannot effectively evaluate the medical capabilities of LLMs (Zhou et al., 2023; Wu et al., 2024). (1) Firstly, existing benchmarks are often based on data collected from online consultation platforms or medical textbooks (Wang et al., 2023), which could easily be included in the training data of LLMs, that is, leading to **data leakage or contamination** and thus biasing the performance evaluation of LLMs. (2) Secondly, the departmental setup in modern medicine is designed to address the complex medical needs of different structures and functions of human organs. The specific skills and treatment methods vary significantly across

Figure 1: The workflow diagram of **ClinicalAgent**. ClinicalAgent covers the entire process starting from the moment a patient enters the clinic and ending when the patient is discharged, which includes six key steps: 1) department guide; 2) preliminary consultation; 3) laboratory examination; 4) imageological examination; 5) final consultation; 6) medical treatment.

different departments. However, existing evaluation benchmarks overlook the characteristics of **multi-departmental and highly specialized nature** of modern medicine, hence they are insufficient in capturing performance differences across departments. (3) Thirdly, existing evaluation methods typically confine themselves to multiple-choice questions (Jin et al., 2020a; Pal et al., 2022; Zhu et al., 2019), which does **not align with real-world clinical diagnostic scenarios**. In actual medical environments, patients seek medical services because they are uncertain about their health conditions, rather than knowing the possible disease options and then seeking a doctor's judgment. Besides, recent studies (Zheng et al., 2023a) have demonstrated that using multiple-choice questions to evaluate LLMs is **not robust and introduces significant biases**. (4) Last but not least, there is currently no evaluation method that can comprehensively and reliably evaluate the **end-to-end practicality** of LLMs in the entire clinical diagnosis process, which starts from the moment a patient enters the clinic and ends when the patient is discharged. This issue will, in turn, limit the design and evaluation of practical medical agents powered by LLMs and harm the exploitation of the full potential of LLMs.

To address these limitations, we introduce **ClinicalBench**, an end-to-end multi-departmental clinical diagnostic evaluation benchmark for **effectively and comprehensively** evaluating the clinical diagnostic capabilities of LLMs. ClinicalBench is based on real cases that cover 24 departments and 150 diseases. ClinicalBench consists of 8 clinical diagnostic tasks. We ensure that ClinicalBench does not have data leakage. We evaluate the clinical diagnostic capabilities of LLMs in two dimensions. The *task* dimension measures the performance of each model in different tasks, while the *department* dimension evaluates the performance difference of each model across various medical specialties. Additionally, we propose four novel metrics (**ClinicalMetrics**) to precisely measure the effectiveness of LLMs in department guide and their clinical diagnostic capabilities. We evaluate 17 mainstream LLMs on ClinicalBench. The evaluation results reveal their performance in different departmental scenarios, reflecting their strengths and weaknesses in simulating human medical practice. We find that **different LLMs typically excel in different departmental areas, and no single LLM can perform excellently well in all departmental domains**. This behavior conforms to the needs of modern medical specialization.

Inspired by the findings, we propose **ClinicalAgent**, a novel clinical diagnostic agent that dynamically allocates the $\mathcal{K}$ most relevant departments and assigns $\mathcal{N}$ clinicians from each department for a collaborative consultation based on the patient's chief complaint. Leveraging flexible department scheduling and clinician allocation strategies, ClinicalAgent delivers a unified diagnostic result. Figure 1 depicts the workflow of ClinicalAgent. Experiments show that ClinicalAgent outperforms top-performing LLMs in clinical diagnostic performance, demonstrating the importance of aligning with modern medical practice for advancing agentic medical systems. We release **ClinicalLab**, a comprehensive clinical diagnosis agent alignment suite, including ClinicalBench, ClinicalMetrics, and ClinicalAgent, to promote development of clinical diagnostic agents. Our contributions can be summarized as follows:

- **ClinicalBench**: We establish the first real-case-based, data-leakage-free, end-to-end multi-departmental benchmark for evaluating the clinical diagnostic capabilities of LLMs. This benchmark covers most departments (totalling 24) and most varieties of diseases (totalling 150).

- **ClinicalMetrics**: We introduce four novel metrics to evaluate the practicality and effectiveness of the results generated by LLMs in real clinical scenarios.

- **ClinicalAgent**: We propose a clinical diagnostic agent tailored for modern medical settings. It surpasses competitive LLMs in the ClinicalBench evaluation and provides comprehensive validation and analysis of its practicality.

Table 1: Comparison between existing benchmarks and datasets for evaluating LLMs in the medical field.

| Benchmark | #Samples | Multi-Departmental | End-to-End | Language | Task Type | Data Source |
|---|---|---|---|---|---|---|
| CMB-Exam | 280,839 | ✓ (Human-annotated) | × | Chinese | Multi-Choice QA | Public (Examinations) |
| CMB-Clin | 74 | × | × | Chinese | Generative QA | Public (Textbooks) |
| CMExam | 68,119 | ✓ (GPT4-annotated) | × | Chinese | Multi-Choice QA | Public (Examinations) |
| MedBench | 1,025 | × | × | Chinese | Multi-Choice QA | Public (Examinations) & Private (Real medical records) |
| MMedBench | 1,136 | ✓ (GPT4-annotated) | × | Six Languages | Multi-Choice QA | Public (Examinations & Websites) |
| MedQA (USMLE-style part) | 1,273 | × | × | English | Multi-Choice QA | Public (Examinations) |
| PubMedQA | 500 | × | × | English | Multi-Choice QA | Public (Websites) |
| MMLU (Six medical subtasks) | 1,089 | × | × | English | Multi-Choice QA | Public (Examinations) |
| HealthSearchQA | 3,173 | × | × | English | Multi-Choice QA | Public (Websites) |
| **ClinicalBench (Ours)** | 1,500 | ✓ (Collected from various departments) | ✓ | Chinese & English | Generative QA | Private (Real medical records) |

## 2 RELATED WORK

### 2.1 EXISTING MEDICAL BENCHMARKS

The Chinese Medical Benchmark (Wang et al., 2023) consists of CMB-Exam and CMB-Clin, with data collected from various public examination databases and multiple-choice questions from textbooks. MedBench (Cai et al., 2024a) also includes multiple-choice questions from the Chinese mainland medical licensing examination and 1,025 QA pairs based on electronic medical records. MMedBench (Qiu et al., 2024) is a multilingual medical evaluation benchmark covering six languages, with data sourced from medical textbooks and open-source medical websites in different languages. Different from our method of collecting real cases from different departments, MMedBench and CMExam (Liu et al., 2023) utilize GPT-4 to categorize each question according to a pre-defined list of departments. However, according to the experimental results shown in Table 3, GPT-4 exhibits significant errors in departmental classification. MedQA (Jin et al., 2020b) consists of multiple-choice questions collected from professional medical board examinations. Its USMLE-style English subset offers four possible answer options for each question and is widely used to evaluate the performance of LLMs in the medical domain. PubMedQA (Jin et al., 2019) is a biomedical QA dataset collected from the PubMed website, where questions need to be answered with Yes/No/Maybe. MMLU (Hendrycks et al., 2021) is a benchmark covering 57 tasks across multiple domains, recent work (Singhal et al., 2022) extracts 6 medicine-related tasks from MMLU to evaluate the medical capabilities of LLMs. Furthermore, HealthSearchQA (Singhal et al., 2022) is a dataset based on medical conditions and related symptoms frequently searched by users on search engines, aiming to evaluate a model's ability to handle real-world user's concerns about their health.

Table 1 compares statistics of existing benchmarks and datasets for evaluating LLMs in the medical field and our ClinicalBench, including the number of samples, task types, language coverage, and data sources. In summary, the main shortcomings of existing evaluation benchmarks include: (1) **lack of end-to-end and evenly distributed departmental coverage** to prevent evaluation bias; (2) **data sources** often come from **easily accessible** online consultation platforms, medical textbooks, and professional examinations, which **poses high risks of data leakage**; (3) existing benchmarks primarily evaluate medical knowledge through **multiple-choice questions, which not only differ significantly from real-world diagnostic scenarios but also lack robustness**.

### 2.2 EXISTING AGENTS FOR MEDICAL APPLICATIONS

Recent works attempt to solve medical and healthcare issues through paradigms of dividing tasks and collaboration among multiple agents. MedAgent (Tang et al., 2023) is the first multi-agent framework in the medical field that improves the accuracy of solving medical multiple-choice questions by allowing the same LLM to play different roles in multi-round collaborative dialogues. However, the design of MedAgent still relies on the multiple-choice question format, which differs significantly from the diagnostic process in the real world. On the other hand, Williams et al. (2023) effectively simulate real-world human behavior patterns through agents played by ChatGPT to address the challenge of incorporating human behavioral factors into epidemic models. MDAgents (Kim et al., 2024) is a multi-agent framework that employs an adaptive decision-making mechanism, addressing medical multiple-choice questions through multiple stages of checking problem complexity, dynamically recruiting experts, reasoning, and decision-making. Meanwhile, Agent Hospital (Li et al., 2024) improves diagnostic accuracy by simulating nearly all medical processes within a hospital and trains doctor agents through doctor-patient interaction simulations. However, due to the lack of a comprehensive dataset covering the entire medical process, currently its effectiveness is only validated on the MedQA multiple-choice dataset. **In summary, existing medical agents suffer from severe limitations and constraints in designs and evaluations due to the lack of benchmarks and**

Figure 2: The data management pipeline for **ClinicalBench**.

**datasets based on real diagnostic processes**. Therefore, developing an evaluation benchmark based on real medical diagnostic processes is critical. Furthermore, most previously proposed medical agents perform clinical diagnosis by using prompts to make the same model play different roles. In contrast, our ClinicalAgent is driven by the evaluation results and analysis of prior models on ClinicalBench. **ClinicalAgent uses different models to play different roles and implements dynamic department scheduling and doctor allocation strategies for clinical diagnosis, ensuring a strong alignment with real hospital settings**.

## 3    CLINICALBENCH: AN END-TO-END, REAL-CASE-BASED, DATA-LEAKAGE-FREE BENCHMARK FOR MULTI-DEPARTMENT CLINICAL DIAGNOSTIC EVALUATION

In this section, we provide a detailed description and analysis of the ClinicalBench benchmark, including data sources and licensing information (Section 3.1), the data collection and quality control processes (Section 3.2), statistical overviews of the relevant datasets (Section 3.3), and descriptions of the 8 medical tasks (Section 3.4).

### 3.1    DATA SOURCES & LICENSES

The data samples used in the ClinicalBench benchmark are sourced from real clinical medical records of officially certified Grade 3A hospitals in China[1]. The collection of this data strictly adheres to the principles of patient privacy protection. No information related to the hospitals is disclosed. As detailed in Data Processing & Quality, to protect patient privacy, any personally identifiable information (PII) of patients, treatment regions, or other sensitive information has been manually identified and removed by the team of doctors. All data is obtained legally and ethically, and has been reviewed and approved by the Ethics Committees of the relevant hospitals, ensuring that research activities on these data comply with ethical and legal obligations. In Appendix A.7, we discuss the certification documents issued by medical institutions and notary offices.

We are committed to responsible data management and strictly follow relevant laws and regulations involving the collection, use, and distribution of protected health information. To ensure the legal and regulated use of the dataset, we have formulated the **ClinicalBench Usage and Data Distribution License Agreement**, which can be found in the supplementary materials. This agreement strictly requires all users to use the data solely for research purposes and to adhere to strict regulations protecting patient privacy, prohibiting any form of personal information tracking or identification. Through these measures, we ensure the legality and ethics of data acquisition and use while supporting research that may promote the development of LLMs in clinical diagnostics. Appendix A.1 provides a detailed explanation of how we securely distribute the dataset.

### 3.2    DATA PROCESSING & QUALITY

The ClinicalBench benchmark is manually created by three senior clinicians and two AI researchers. As shown in Figure 2, the creation process covers 4 key steps, as follows. (1) The **Data collection** step focuses on **authenticity, diversity, privacy**. Based on department divisions and common disease types in each department, the medical team selects representative real cases for each disease from the hospital case database with permission for research. Given that these clinical case data is the **private** information of hospitals, the risk of data leakage to any LLMs is completely eliminated. (2) The **Professional Knowledge Review** step ensures the **accuracy** of the data. The team of doctors conducts a detailed professional review of the diagnostic information, treatment process, and results

---

[1]Grade 3A hospitals are the highest level hospitals in China's "three-grade, six-class" classification system.

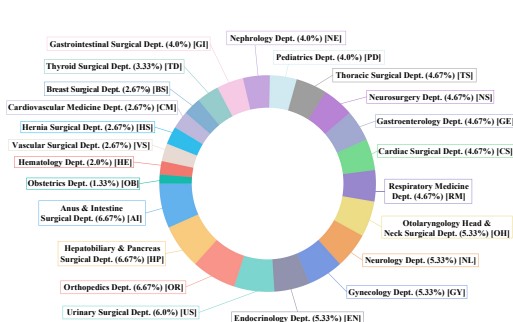
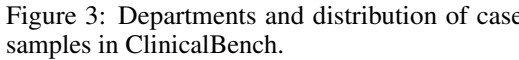

Figure 3: Departments and distribution of case samples in ClinicalBench.

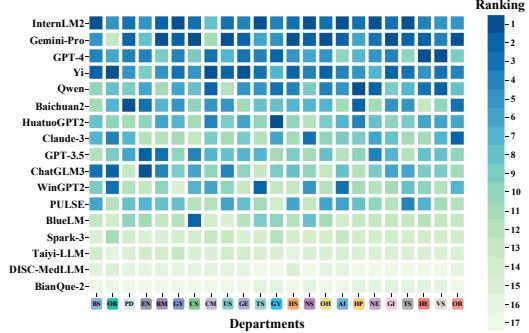

Figure 4: Ranking of different LLMs across departments, with the x-axis representing department abbreviations (abbreviations correspond to Figure 3) and the y-axis representing the models.[3]

of each case to ensure the medical accuracy and proficiency of the data. (3) The **Privacy Protection and De-identification** step ensures **privacy protection**. To protect patient privacy, the team of doctors conducts two rounds of independent reviews to identify and remove any content that could reveal patient identities, treatment regions, or other sensitive information. (4) The **Data Integrity and Compliance Check** step aims for **completeness and ethical compliance**. Two AI researchers are responsible for reviewing the data to ensure that each record is complete, and meets the medical task requirements set in Section 3.4. Additionally, they reconfirm that the dataset does not contain any sensitive information and strictly complies with the ethical guidelines.

## 3.3 DATA STATISTICS

ClinicalBench is a fine-grained evaluation benchmark based on chain-of-thought reasoning, specifically designed for multi-departmental clinical diagnosis, covering 24 departments such as pediatrics, orthopedics, and neurosurgery. Figure 3 presents detailed information about the various departments covered by ClinicalBench. It involves 150 different diseases, each comprising 10 specific cases, totaling 1500 samples, with an average of about 1000 tokens per case[2]. Table 1 provides relevant information about ClinicalBench. **To the best of our knowledge, ClinicalBench is the most comprehensive clinical diagnostics evaluation benchmark to date, covering the widest range of departments and diseases**.

Each case $\mathcal{E}$ in ClinicalBench contains detailed clinical data, such as the patient's gender, age[4], chief complaint $\mathcal{CC}$, medical history $\mathcal{MH}$, physical examination $\mathcal{PE}$. Additionally, the cases include various medical imaging reports $\mathcal{IR}$, such as X-rays, computed tomography (CT) scans, magnetic resonance imaging (MRI), and ultrasound examinations, as well as biochemical, immunological, microbiological, and pathological laboratory examination reports $\mathcal{LR}$ from biological samples such as blood, urine, and cerebrospinal fluid. The preliminary diagnosis, diagnostic basis, differential diagnosis, and treatment plans provided by doctors from each department collectively support a range of end-to-end clinical decision-making processes. A complete data example is provided in Table 5-9.

## 3.4 TASK OVERVIEW

ClinicalBench systematically evaluates the end-to-end practicality of LLMs in clinical diagnosis by simulating the complete patient visit process, from the patient's entry into the hospital to their discharge. We divide the entire process into 8 specific tasks, covering various stages from preliminary reception to final diagnosis and treatment plan formulation. These tasks are categorized into three main functional groups: **department guide**, **clinical diagnosis**, and **imaging diagnosis**.

---

[2]All samples are originally in Chinese and translated into English using GPT-4 for reading and usage.

[3]The ranking of each department is determined based on the **Avg. Score** metric described in Table 2.

[4]Gender and age are crucial factors in clinical diagnosis, and with PII and other sensitive information removed, they alone are typically not considered PII. Therefore, we decide to retain these two information.

### 3.4.1 DEPARTMENT GUIDE (MULTI-CHOICE QA WITH 24 OPTIONS)

When patients first enter the hospital, guiding them to the correct department for further consultation, examination and treatment is crucial for providing targeted medical services. The **Department Guide (DG)** task requires the model to choose the most reasonable department from a given set of 24 departments based on a $\mathcal{CC}$, i.e., $\mathcal{DP} = \mathrm{DG}(\mathcal{CC})$, where $\mathcal{DP}$ denotes the selected department. This task evaluates the decision-making accuracy, and instruction-following capability of LLMs in real-world medical scenarios.

### 3.4.2 CLINICAL DIAGNOSIS (GENERATIVE QA)

The Clinical Diagnosis task evaluates the LLM's clinical diagnostic capabilities in real cases and its analysis and interpretation abilities. Clinical Diagnosis includes 6 sub-tasks, each designed according to the standard diagnostic process recommended by the Chinese official record (NHC, 2022) and using the chain-of-thought pattern. The 6 sub-tasks are defined as follows.

**Preliminary Diagnosis (PD)**: The model generates a list of possible diseases $\mathcal{P}$ based on $\mathcal{CC}$, $\mathcal{MH}$, and $\mathcal{PE}$, i.e., $\mathcal{P} = \mathrm{PD}(\mathcal{CC}, \mathcal{MH}, \mathcal{PE})$. This task primarily evaluates the model's information synthesis and reasoning abilities.

**Diagnostic Basis (DB)**: The model needs to provide supportive medical evidence $\mathcal{B}$ for each possible disease in $\mathcal{P}$, i.e., $\mathcal{B} = \mathrm{DB}(\mathcal{E}, \mathcal{P})$. This task mainly evaluates the model's evidence extraction ability and helps mitigate hallucinations.

**Differential Diagnosis (DD)**: After considering $\mathcal{B}$, the model should perform a differential analysis $\mathcal{D}$ to exclude other diseases that have different causes but similar manifestations, i.e., $\mathcal{D} = \mathrm{DD}(\mathcal{E}, \mathcal{P}, \mathcal{B})$. This task primarily evaluates the model's abilities in analytical comparison and decision-making.

**Final Diagnosis (FD)**: Integrating all information, the model needs to determine the final diagnosis $\mathcal{F}$, i.e., $\mathcal{F} = \mathrm{FD}(\mathcal{E}, \mathcal{P}, \mathcal{B}, \mathcal{D})$. This task mainly evaluates the model's comprehensive judgment ability and the accuracy of clinical diagnosis.

**Principle of Treatment (PT)**: The model determines the treatment principles and guidelines $\mathcal{G}$ for $\mathcal{F}$, i.e., $\mathcal{G} = \mathrm{PT}(\mathcal{F})$. This task primarily evaluates the model's memorization of medical standard operating procedures and knowledge points.

**Treatment Plan (TP)**: The model formulates specific treatment steps and plans $\mathcal{T}$ for the given case $\mathcal{E}$, including medication, surgical intervention, and physical therapy, i.e., $\mathcal{T} = \mathrm{TP}(\mathcal{E}, \mathcal{P}, \mathcal{B}, \mathcal{D}, \mathcal{F}, \mathcal{G})$. This task mainly evaluates the model's abilities in knowledge application and strategic planning.

### 3.4.3 IMAGING DIAGNOSIS (GENERATIVE QA)

The **Imaging Diagnosis (ID)** task requires LLMs to understand detailed textual reports of medical images, identify key features of lesions, such as tumors, inflammation, or other pathological changes, and provide imaging diagnosis results $\mathcal{IDR}$, i.e., $\mathcal{IDR} = \mathrm{ID}(\mathcal{IR})$. This task evaluates the LLM's ability to analyze and interpret medical imaging reports for diagnostic support.

## 4 EXPERIMENTS OF LLMS ON CLINICALBENCH

### 4.1 MODELS

To comprehensively analyze strengths and weaknesses of a broad selection of competitive LLMs on different tasks and departments in ClinicalBench, we evaluate **11 general LLMs**, including GPT-4 (OpenAI, 2023), GPT-3.5 (OpenAI, 2023), Gemini-Pro-1.0 (Gemini, 2023), InternLM2-20B-Chat (Cai et al., 2024b), Yi-34B-Chat (Young et al., 2024), Qwen-72B-Chat (Bai et al., 2023), Baichuan2-13B-Chat (Yang et al., 2023), Claude3 (Anthropic, 2024), ChatGLM3-6B (Du et al., 2021), BlueLM-7B-Chat (Team, 2023), and Spark-3 (Iflytek Co., 2024). We also evaluate **6 Chinese medical LLMs**, including HuatuoGPT2-34B (Chen et al., 2023a), WiNGPT2-14B-Chat (Research, 2024), PULSE-20B (Zhang et al., 2023), Taiyi-LLM (Luo et al., 2023), DISC-MedLLM(Bao et al., 2023), and BianQue-2 (Chen et al., 2023b). To ensure the reproducibility of the experiments, we use greedy decoding for each LLM (set the temperature parameter to 0 or set the do_sample parameter to *False*). The experimental evaluation is completed through API calls and 8 NVIDIA A6000 GPUs.

## 4.2 EVALUATION METRICS (CLINICALMETRICS)

We propose four novel metrics to precisely measure the effectiveness of LLMs in department guide and their clinical diagnostic capabilities, as follows. Additionally, we use metrics such as accuracy, BLEU (Papineni et al., 2002), ROUGE (Lin & Hovy, 2003), and BertScore (Zhang et al., 2020) to evaluate the experimental results. For more details and **the handling of medical synonyms**, please refer to Appendix A.2.

**Department Win Rate (DWR)** measures the relative performance of LLMs in clinical diagnosis across different medical departments, which is calculated as Eq. 1,

$$\text{DWR} = \frac{1}{m} \sum_{j=1}^{m} \left( \frac{1}{n} \sum_{i=1}^{n} \left( n - (\mathsf{r}_i^j - 1) \right) \right) \tag{1}$$

where $m$ and $n$ denote the total number of departments and models, respectively; $r_i^j$ is the ranking of model $i$ in department $j$ (based on the Avg. score).

**Department Instruction Following Rate (DIFR)** measures the degree to which LLMs follow task instructions when generating the quantity and names of ranked departments. It is the average of two sub-metrics: DIFR-Q and DIFR-N.

Department Quantity Following Rate (DIFR-Q) measures the consistency between the quantity of top $K'$ departments generated by the model and the quantity of $K$ departments specified in the task instructions, which is calculated as Eq. 2,

$$\text{DIFR-Q} = \frac{\sum_{i=1}^{N} \mathbf{S}_{\{K_i'=K\}}}{N} \tag{2}$$

where $\mathbf{S}_{\{K_i'=K\}}$ is an indicator function that equals 1 if the quantity of $K_i'$ departments generated for the $i$-th sample, matches the specified quantity $K$ in the task instructions, and 0 otherwise. $N$ is the total quantity of test samples.

Department Name Following Rate (DIFR-N) measures the consistency of the department names generated by the model with the names in a predefined list of 24 departments, which is calculated as Eq. 3,

$$\text{DIFR-N} = \frac{\sum_{i=1}^{N} \sum_{j=1}^{K_i'} \text{match}(i,j)}{\sum_{i=1}^{N} K_i'} \tag{3}$$

where $K_i'$ represents the number of departments generated for the $i$-th sample, and the match function $\text{match}(i,j)$ checks whether the department name generated for the $j$-th position in the $i$-th sample is in the predefined list of departments. If so, $\text{match}(i,j) = 1$; otherwise, it is 0. $N$ is the total number of test samples.

**Comprehensive Diagnostic Rate (CDR)** measures the accuracy of LLMs in simultaneously predicting correct department guide and disease diagnosis, which is calculated as Eq. 4,

$$\text{CDR} = \frac{\sum_{i=1}^{N} \text{S}_{\text{guide}}(i) \times \text{S}_{\text{diagnosis}}(i)}{N} \tag{4}$$

where $\text{S}_{\text{guide}}(i)$ and $\text{S}_{\text{diagnosis}}(i)$ are indicator functions of correctness for the $i$-th sample in the guide and diagnosis tasks, respectively. If the prediction is correct, the function value is 1, and 0 otherwise. $N$ is the total number of test samples.

**Acceptability** measures the comprehensive performance of LLMs in terms of prediction accuracy in department guide and disease diagnosis tasks, as well as the language quality of other generated diagnostic content, reflecting the overall effectiveness of the model in handling complex clinical situations. It is calculated as Eq. 5,

$$\text{Acceptability} = \left( \frac{1}{N} \sum_{i=1}^{N} \text{CDR}(i) \right) \times \left( \frac{1}{NKM} \sum_{i=1}^{N} \sum_{k=1}^{K} \sum_{m=1}^{M} \text{Metric}_m^k(i) \right) \tag{5}$$

where $\text{CDR}(i)$ is the indicator function of the $i$-th sample correctly predicting both guide and diagnosis tasks. $\text{Metric}_m^k(i)$ represents the score of the $m$-th evaluation metric for the $i$-th sample in the $k$-th diagnostic task. $N$ is the total number of test samples. $M$ is the number of evaluation metrics, which include BLEU, ROUGE, and BertScore. $K$ is the total number of diagnostic tasks, including diagnostic basis, differential diagnosis, principles of treatment, treatment plans, and imaging diagnosis.

Table 2: Performance of LLMs across 8 medical tasks in ClinicalBench. The PD and FD tasks are evaluated based on Accuracy, while other Clinical Diagnosis subtasks are evaluated using the average of of BLEU, ROUGE, and BertScore scores. For an LLM, we compute the average (**Avg.**) score over all the 9 scores under the 8 tasks as its overall performance on ClinicalBench.

| Model | Department Guide | | Clinical Diagnosis | | | | | | Imaging Diagnosis | Overall | | | |
|---|---|---|---|---|---|---|---|---|---|---|---|---|---|
| | Acc | DIFR | PD | DB | DD | FD | PT | TP | | DWR | CDR | Acceptability | Avg. |
| InternLM2-20B-Chat | **64.47** | 97.18 | **78.20** | 46.22 | 30.98 | 51.13 | 33.05 | 30.75 | 35.88 | **14.91** | 31.40 | 11.11 | **51.98** |
| Gemini-Pro | 62.07 | 94.45 | 69.13 | **50.59** | 31.26 | 50.00 | 32.54 | 29.29 | 36.35 | 14.48 | 30.13 | 10.85 | 50.63 |
| Yi-34B-Chat | 58.80 | **98.08** | 72.60 | 47.41 | **31.74** | 48.33 | 32.86 | 28.62 | 36.34 | 14.30 | 26.13 | 9.25 | 50.53 |
| GPT-4 | 61.47 | 87.02 | 71.47 | 40.94 | 30.69 | **54.20** | 32.52 | 29.10 | 37.61 | 12.70 | **33.27** | **11.37** | 49.45 |
| Qwen-72B-Chat | 63.67 | 85.35 | 73.33 | 39.72 | 30.05 | 53.33 | 31.39 | 29.46 | 34.00 | 11.96 | 33.07 | 10.89 | 48.92 |
| HuatuoGPT2-34B | 59.20 | 97.37 | 70.60 | 42.58 | 26.35 | 47.67 | 32.20 | 29.44 | 30.40 | 10.17 | 27.13 | 8.73 | 48.42 |
| Baichuan2-13B-Chat | 53.73 | 87.43 | 74.00 | 43.96 | 28.10 | 44.80 | 31.53 | 30.19 | 36.91 | 10.96 | 21.53 | 7.35 | 47.85 |
| Claude3 | 58.93 | 92.47 | 62.93 | 33.76 | 26.32 | 49.93 | **34.12** | **31.25** | 35.26 | 9.30 | 27.27 | 8.76 | 47.22 |
| GPT-3.5 | 56.80 | 89.50 | 57.27 | 38.66 | 28.24 | 47.13 | 29.12 | 30.78 | 37.78 | 9.96 | 24.27 | 7.99 | 46.14 |
| ChatGLM3-6B | 46.40 | 95.88 | 58.00 | 42.51 | 27.83 | 38.40 | 31.78 | 30.88 | 35.02 | 9.96 | 15.93 | 5.35 | 45.19 |
| WiNGPT2-14B-Chat | 42.13 | 79.15 | 57.33 | 43.74 | 24.61 | 46.87 | 27.86 | 25.74 | **41.39** | 8.57 | 16.53 | 5.40 | 43.20 |
| PULSE-20B | 54.93 | 50.08 | 49.20 | 37.89 | 26.71 | 41.53 | 31.15 | 30.34 | 37.07 | 9.00 | 20.87 | 6.81 | 39.88 |
| BlueLM-7B-Chat | 45.33 | 82.13 | 44.67 | 35.24 | 18.78 | 36.00 | 28.84 | 26.72 | 34.40 | 6.26 | 16.73 | 4.82 | 39.12 |
| Spark-3 | 34.87 | 95.7 | 60.53 | 36.88 | 24.44 | 35.07 | 9.77 | 8.37 | 31.63 | 4.17 | 11.80 | 2.62 | 37.47 |
| Taiyi-LLM | 44.00 | 97.07 | 16.87 | 17.43 | 10.79 | 18.27 | 12.17 | 11.32 | 31.65 | 3.22 | 7.60 | 1.27 | 28.84 |
| DISC-MedLLM | 45.67 | 73.59 | 3.27 | 1.84 | 1.67 | 2.07 | 1.90 | 1.67 | 20.63 | 1.70 | 0.40 | 0.02 | 12.87 |
| BianQue-2 | 0.07 | 10.21 | 2.67 | 1.74 | 0.04 | 0.00 | 3.64 | 4.73 | 29.51 | 1.39 | 0.00 | 0.00 | 4.50 |

## 4.3 RESULTS & ANALYSIS

Table 2 shows the performance of 17 LLMs on ClinicalBench. Note that all experiments in this work are conducted on the original Chinese samples of ClinicalBench[5].

**Performance on each task. On the department guide task**, InternLM2 performs the best in guiding patients to the correct department (Accuracy). We find LLMs occasionally hallucinate and incorrectly guide patients to non-existent departments. Yi-Chat excels at following instructions, generating department names with high accuracy and minimal hallucinations. **Among clinical diagnosis sub-tasks**, InternLM2 leads in the PD task, demonstrating a strong ability to identify possible diseases based on initial symptoms. Gemini-Pro performs the best in the DB task, effectively extracting supportive medical evidence and exhibiting good reliability in clinical settings. Yi-Chat performs the best in the DD task, effectively eliminating interference from similar conditions with fewer hallucinations. GPT-4 excels in the FD task, demonstrating superior judgment and diagnostic accuracy. Claude3 leads in the PT and TP tasks, showcasing its deep understanding and application of basic medical knowledge. **On the imaging diagnosis task**, WiNGPT2 is significantly ahead all other models, indicating strong capabilities in analyzing medical imaging reports and utilizing medical equipment information.

**Overall Performance.** InternLM2 achieves the top DWR of 14.91 and Avg. 51.98, indicating its superiority over other models across multiple departments with balanced performance. We hypothesize that the diverse pre-training data, thorough data filtering methods, and careful executions of pre-training, supervised fine-tuning, and alignments through a novel reinforcement learning (Cai et al., 2024b) may have contributed to the superior performance of InternLM2. GPT-4 performs the best in simultaneously predicting the correct department and disease diagnosis (CDR), showing its ability to complete 33.27% of end-to-end medical consultations. Moreover, GPT-4 demonstrates the effectiveness of its content quality and accuracy with a top acceptability score of 11.37.

> **Overall Findings**
>
> From the *task* dimension, we observe that **existing LLMs are not yet fully capable of handling end-to-end clinical diagnosis tasks**. Interestingly, we find that **the clinical diagnostic capabilities of general LLMs are better than those of the specialized medical LLMs**; hence, how to improve specialized models with medical data remains an open challenge.
>
> From the *department* dimension, Figure 4 shows the rankings of different models across various departments, demonstrating that **no single LLM excels in all departmental domains**. This finding aligns well with the reality of human medical experts, where no single expert can master all departmental domains.

[5]Due to high costs, we are unable to conduct these repetitive experiments, but we provide code to support the evaluation using the English version of ClinicalBench.

Table 3: Detailed performance of LLMs on the Departmental Guide task.

| Model | Acc@1 | Acc@3 | Acc@5 | DIFR-Q | DIFR-N | Avg. |
|---|---|---|---|---|---|---|
| Gemini-Pro | 62.07 | 85.73 | **92.00** | **100.0** | 88.90 | 85.74 |
| Yi-Chat | 58.80 | 83.60 | 86.07 | 99.93 | **96.23** | 84.93 |
| InternLM2 | **64.47** | 79.87 | 84.67 | **100.0** | 94.37 | 84.68 |
| HuatuoGPT2 | 59.20 | 80.47 | 88.87 | **100.0** | 94.73 | 84.65 |
| GPT-4 | 61.47 | **86.13** | 90.13 | **100.0** | 74.03 | 82.35 |
| Claude3 | 58.93 | 79.33 | 85.67 | **100.0** | 84.93 | 81.77 |
| GPT-3.5 | 56.80 | 81.13 | 89.00 | **100.0** | 79.00 | 81.87 |
| Qwen | 63.67 | 80.73 | 86.67 | **100.0** | 70.70 | 80.35 |

Table 4: Performance of ClinicalAgent and top-performing LLMs using three evaluation methods.

| Model | Automatic Score | | | | Human Score | GPT-4o Score |
|---|---|---|---|---|---|---|
| | DWR | CDR | Acceptability | Avg. | | |
| InternLM2 | 14.91 | 31.40 | 11.11 | 51.98 | 54.66 | 85.09 |
| GPT-4 | 12.70 | 33.27 | 11.37 | 49.45 | 55.90 | **90.93** |
| Gemini-Pro | 14.48 | 30.13 | 10.85 | 50.63 | 58.84 | 79.61 |
| Yi-Chat | 14.30 | 26.13 | 9.25 | 50.53 | 52.42 | 86.61 |
| Agent#3@1 | 14.30 | 52.73 | 17.82 | **54.46** | 59.42 | 85.61 |
| Agent#1@3 | 14.30 | **54.00** | **18.22** | 51.66 | **62.76** | 89.40 |
| Agent#1@1 | **17.00** | 35.13 | 12.57 | 53.02 | 57.84 | 82.46 |

# 5 CLINICALAGENT: AN END-TO-END CLINICAL AGENT ALIGNED WITH REAL-WORLD MULTI-DEPARTMENTAL CLINICAL DIAGNOSTIC PRACTICES

## 5.1 CLINICALAGENT: METHODOLOGY

Our findings in Section 4.3, shows obvious performance deficiencies when requiring a *single* model to play different departmental roles in highly specialized medical scenarios. We believe that setting specific agents for different departments, i.e., the agent for each department is powered by the LLM that performs best in the specific domain, aligns better with the real-world practices of human medical experts. Therefore, we propose **ClinicalAgent, an End-to-End Clinical Agent Aligned with Real-World Multi-Departmental Clinical Diagnostic Practices**, with its diagnostic process shown in Figure 1. ClinicalAgent operates in 6 stages, as follows. Details of the stages and the workflow algorithm can be found in Appendix A.3.

**Departmental Guide**: The patient $\mathcal{PA}$ presents a chief complaint $\mathcal{CC}$ to the patient navigator agent $\mathcal{PN}$. Based on $\mathcal{CC}$ and the prior knowledge $\mathcal{PK}$ of LLM rankings obtained in Table 2, agent $\mathcal{PN}$ arranges $\mathcal{N}$ clinicians from the $\mathcal{K}$ most relevant departments in the consultation room to prepare a preliminary consultation for $\mathcal{PA}$. The purpose of this stage is to quickly and accurately identify the patient's initial needs and arrange for the appropriate specialty medical team to diagnose.

**Preliminary Consultation**: The $\mathcal{PA}$ consults with multiple clinicians from different departments in the consultation room, such as pediatrics clinician powered by GPT-4, orthopedic clinician powered by InternLM2, and hematology clinician powered by Gemini-Pro. Each clinician makes a preliminary diagnosis $\mathcal{P}$ based on the $\mathcal{CC}$, $\mathcal{MH}$, and $\mathcal{PE}$, and decide on further laboratory tests and imageological examinations.

**Laboratory Examination**: According to the preliminary consultation advice, the $\mathcal{PA}$ proceeds to the biochemistry laboratory for a series of necessary examinations, such as blood tests and urine analysis. Subsequently, based on the laboratory report $\mathcal{LR}$ generated by the medical instruments, the biochemist provides the laboratory diagnostic results $\mathcal{LDR}$.

**Imageological Examination**: According to the preliminary consultation advice, the $\mathcal{PA}$ undergoes necessary imageological examinations in the radiology room, such as X-rays, CT scans, or MRI. Subsequently, the radiologist provides an imaging diagnosis result $\mathcal{IDR}$ based on the radiological reports $\mathcal{IR}$ of these medical images.

**Final Consultation**: The $\mathcal{PA}$ returns to the consultation room and provides $\mathcal{LDR}$ and $\mathcal{IDR}$ to the medical team. Based on the newly acquired reference information, the medical team conducts a step-by-step analysis and ultimately provides results including $\mathcal{B}$, $\mathcal{D}$, $\mathcal{F}$, $\mathcal{G}$, and $\mathcal{T}$.

**Medical Treatment**: The $\mathcal{PA}$ receives corresponding treatment based on $\mathcal{G}$ and $\mathcal{T}$, including treatments conducted in the treatment room and medication dispensed in the pharmacy. Finally, the patient leaves the hospital after completing the treatment.

## 5.2 CLINICALAGENT: EVALUATIONS

**Experimental Setup** As described in Section 2.2, existing medical agents such as MedAgent (Tang et al., 2023) only support multiple-choice questions and do not support end-to-end diagnosis. Therefore, we compare the default configuration of ClinicalAgent with several variants and the top-performing LLMs in Table 2 to verify the effectiveness of the ClinicalAgent approach. Each variant, Agent#K@N, employs different department scheduling and clinician allocation strategies for clinical diagnosis, where K denotes the number of departments to be scheduled, and N denotes the number of the top clinicians assigned to each department. For example, Agent#1@3 schedules three top clinicians from the same department for diagnosis. We use three evaluation methods to measure the quality and accuracy of the agent's diagnostic results: **automatic evaluation**, **human evaluation**,

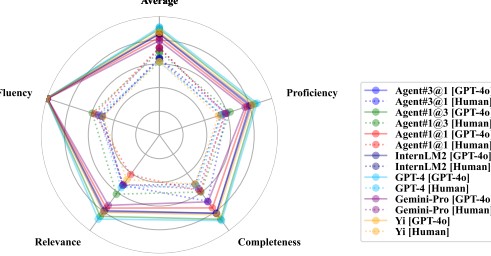

Figure 5: Performance trends of ClinicalAgent and LLMs using different evaluation metrics.

Figure 6: Performance of ClinicalAgent and LLMs on fluency, relevance, completeness, and proficiency as evaluated by human and GPT-4o.

and **GPT-4o evaluation**. A detailed description of the three evaluation methods can be found in Appendix A.4.

According to modern medical practice, accurately guiding patients to the most appropriate departments is a crucial initial step in ClinicalAgent. Therefore, we thoroughly evaluate the patient navigation capabilities of various LLMs to select the best patient navigator. As shown in Table 3, despite the fact that Yi-Chat yields slightly lower Acc@1 than other models, we choose it as the patient navigator due to its near-perfect instruction following performance and minimal hallucinations. Detailed analysis and explanation of the reasons can be found in Appendix A.5.

**Results & Analysis** Table 4 and Figure 5 illustrate the performance of 4 LLMs and 3 configurations of ClinicalAgent using the three different evaluation methods. Tables 10-31 in the Appendix provide detailed case studies, including case information, diagnostic results from human doctors and seven models. Using both **Automatic Evaluation** and **Human Evaluation** methods, ClinicalAgent (Agent#K@N) based on department scheduling and doctor allocation strategies leads top-performing LLMs with a large margin. With **GPT-4o evaluation**, GPT-4 achieves the highest score of 90.93, which is probably attributable to the evaluation model's preference for its own generated answers (Zheng et al., 2023b). Even so, ClinicalAgent (Agent#K@N) receives competitive GPT-4o scores up to 89.40. Overall, ClinicalAgent (Agent#K@N) excel in handling complex diagnosis tasks, showcasing the effectiveness of collaborative diagnostic strategies across multiple departments and doctors.

> **Overall Findings**
>
> Interestingly, **different configurations of ClinicalAgent exhibit varying performance**. Overall, Agent#1@3 performs the best. We conduct deeper analysis of these variants. Figure 6 shows that Agent#1@3 excels in fluency, relevance, and proficiency in medicine, benefiting from *integrating diagnostic opinions from multiple doctors within the same department*, which effectively fills in gaps and reduces misdiagnosis. However, due to its focus on a single department, Agent#1@3 falls slightly short in completeness and **is more suitable for clinical diagnosis of a single disease or multiple closely related diseases**. Figure 6 shows that Agent#3@1 excels in comprehensiveness and proficiency in medicine, but its fluency and relevance are slightly compromised. This may be attributed to errors in Department Guide Acc@3, leading to noisy opinions from doctors from irrelevant department. Overall, Agent#3@1 benefits from *diverse expertise across departments* and generates high-quality diagnostic results **more suitable for joint clinical diagnosis of multiple complex or loosely related diseases**. In comparison, **Agent#1@1, as an ablation study of Agent#3@1 and Agent#1@3**, while not performing as well as the collaboration of multiple experts, still clearly outperforms other top-performing LLMs, demonstrating **the effectiveness of flexible and optimized department scheduling and doctor allocation strategies in clinical diagnostic agents**.

Notably, as shown in Figure 5, the trends based on different metrics do not completely align. This discrepancy arises from the varying emphases of each metric and highlights the limitations of existing metrics, demanding future studies.

## 6 CONCLUSION

We introduce ClinicalLab, which provides a comprehensive set of resources, tools, and foundational design methodologies for medical agents, offering crucial support for evaluating and designing future LLMs and medical agents. We also provide detailed discussions of **limitations** and **ethical considerations** in Appendix A.6 and A.7.

# 7 REPRODUCIBILITY STATEMENT

1. If you ran experiments (e.g. for benchmarks)...

   (a) Did you include the code, data, and instructions needed to reproduce the main experimental results (either in the supplemental material or as a URL)?
   **Answer**: We add all the data, code, and LLM inference results in the supplementary material, which makes it easy to reproduce our work.

   (b) Did you specify all the training details (e.g., data splits, hyperparameters, how they were chosen)?
   **Answer**: We added a readme and comments in the code in the supplementary material to show all the training details, and explained the parameter settings in Section 4.1 of the main paper.

   (c) Did you report error bars (e.g., with respect to the random seed after running experiments multiple times)?
   **Answer**: To ensure the reproducibility of the experiments, we set the temperature parameter to 0 or set the do_sample parameter to $False$. It is easy to reproduce our results by just running our code. We also added all the inference results in the supplementary files.

   (d) Did you include the total amount of compute and the type of resources used (e.g., type of GPUs, internal cluster, or cloud provider)?
   **Answer**: We report the total amount of compute and the types of resources used in Section 4.1 of the main paper.

2. If you are using existing assets (e.g., code, data, models) or curating/releasing new assets...

   (a) If your work uses existing assets, did you cite the creators?
   **Answer**: Whenever I use existing assets, I make sure to properly cite them and acknowledge the sources in the appropriate places to ensure transparency and fairness, and to respect others' work.

   (b) Did you mention the license of the assets?
   **Answer**: All assets I use are publicly available and comply with the corresponding license agreements.

   (c) Did you include any new assets either in the supplemental material or as a URL?
   **Answer**: We added a completely new evaluation dataset.

   (d) Did you discuss whether and how consent was obtained from people whose data you're using/curating?
   **Answer**: This is the most critical part of the paper and the issue we are most concerned about. We not only obtained official permission from the hospital, but also passed the review of the hospital's ethics review committee. These have been officially notarized and all the certificates can be found in the attachment. In Sections 3.1 and A.7, we discuss these issues in detail.

   (e) Did you discuss whether the data you are using/curating contains personally identifiable information or offensive content?
   **Answer**: This is also the most critical part of the paper and the issue we are most concerned about. We explain these issues in detail in Section 3.2 of the paper. We can guarantee that the data used in ClinicalBench contains all personally identifiable information and cannot identify and track personal information. At the same time, we have established a "ClinicalBench Usage and Data Distribution License Agreement" for the data, and we obtained approval from the Medical Ethics Review Board when acquiring the data

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

# A APPENDIX

## A.1 THE SECURE DISTRIBUTION OF CLINICALBENCH

To further strengthen data security, we adopt an application-based data distribution system. Applicants must review and agree to the terms and commit to maintaining confidentiality. Additionally, we utilize an automated algorithm to embed personalized watermarks in each dataset, based on the applicant's information. This enables us to swiftly trace the source in the event of a data breach. These measures ensure that ClinicalBench can provide valuable data to the medical AI research community while safeguarding privacy.

As of September 30, 2024, we have successfully distributed ClinicalBench datasets 63 times to researchers from various countries and regions, including the United States, China, the United Kingdom, Germany, and Singapore. This reflects the positive impact ClinicalBench is already making in the AI research community and its contribution to global advancements in medical AI. We are confident that ClinicalBench will continue to play a pivotal role in driving progress in medical AI research.

## A.2 EVALUATION METRICS

**Accuracy** indicates the proportion of samples correctly classified by the department to the total number of samples. **BLEU** (Papineni et al., 2002) measures the degree of overlap between the predicted output and the reference output, ranging from 1-gram to 4-gram. **Rouge** (Lin & Hovy, 2003) primarily evaluates based on the longest common subsequence found between the generated texts and the human-written reference ones. **BertScore** (Zhang et al., 2020) leverages a pre-trained BERT model and its deep contextual representations to gauge the semantic similarity between the generated and reference texts. In the experiment, we use the "bert-base-chinese"[6] model along with the "jieba"[7] Chinese segmentation tool to ensure accurate word tokenization.

To ensure the accuracy of BLEU, ROUGE, and BertScore when evaluating medical synonyms, we implement several key steps. First, we compile a list of synonyms for 150 diseases from medical textbooks and online encyclopedias, encompassing a range of expression variations. To address discrepancies in terminology arising from differences in model training data, we also request that 17 models independently generate synonym lists for each disease. We then merge the synonyms collected from textbooks and encyclopedias with those generated by the models. This combined list is rigorously screened by three medical experts among the authors to produce a standardized synonym list.

---

[6]https://huggingface.co/google-bert/bert-base-chinese
[7]https://pypi.org/project/jieba/

Before calculating BLEU, ROUGE, and BertScore, we use this standardized list to align the model-generated terms with the ground truth, minimizing errors due to synonym variations and ensuring the accuracy of the metrics. Furthermore, when distributing the dataset, we include the synonym list to enable other researchers to replicate and validate our work.

### A.3 CLINICALAGENT ALGORITHM

The workflow of the aforementioned ClinicalAgent is shown in Algorithm 1.

**Departmental Guide**: When patient $\mathcal{PA}$ enters the hospital, they submit a simple chief complaint $\mathcal{CC}$ to the patient navigator agent supported by Yi-Chat (Young et al., 2024). The navigator agent, based on the $\mathcal{CC}$ and prior knowledge $\mathcal{PK}$ from Table 2, dynamically arranges the most relevant $\mathcal{K}$ departments $\mathcal{DP}_k (k = 1, 2, \ldots, \mathcal{K})$ according to the complexity of the $\mathcal{CC}$, and assigns $\mathcal{N}$ top clinicians $\mathcal{CL}_n^{\mathcal{DP}_k} (n = 1, 2, \ldots, \mathcal{N})$ from each department $\mathcal{DP}_k$ to form the clinician team $\mathcal{CT}$. The $\mathcal{CT}$ prepares for an initial consultation with $\mathcal{PA}$ in the consultation room. The purpose of this stage is to quickly and accurately identify the patient's initial needs and arrange the appropriate specialized medical team for diagnosis.

**Preliminary Consultation**: First, a lead clinician $\mathcal{CL}_{chair}$ is selected from the clinical team $\mathcal{CT}$ to coordinate and summarize the entire diagnostic process. In the consultation room, the $\mathcal{PA}$ meets simultaneously with multiple clinicians from various departments, such as a pediatrics clinician $\mathcal{CL}_{GPT-4}^{Pediatrics}$ powered by GPT-4, an orthopedic clinician $\mathcal{CL}_{InternLM2}^{Orthopedics}$ powered by InternLM2, and a hematology clinician $\mathcal{CL}_{Gemini-Pro}^{Hematology}$ powered by Gemini-Pro. Each clinician $\mathcal{CL}_i$ briefly reviews the patient's chief complaint $\mathcal{CC}$, medical history $\mathcal{MH}$, and physical examination $\mathcal{PE}$, then independently formulates a preliminary diagnosis $\mathcal{P}_i$ and recommends any necessary laboratory tests $\mathcal{LTS}_i$ or imaging examinations $\mathcal{ITS}_i$. Finally, the $\mathcal{CL}_{chair}$ consolidates all evaluations, synthesizing the diagnostic hypotheses and test suggestions into a unified preliminary diagnosis $\mathcal{P}$, and finalizes the required laboratory tests $\mathcal{LTS}$ and imaging examinations $\mathcal{ITS}$ for $\mathcal{PA}$.

**Laboratory Examination**: Based on the preliminary consultation advice, the $\mathcal{PA}$ proceeds to the biochemical testing room to undergo a series of necessary tests according to the $\mathcal{LTS}$, such as blood tests and urine analysis. Then, based on the laboratory reports $\mathcal{LR}_i$ generated by medical instruments, the biochemist provides the corresponding laboratory diagnostic results $\mathcal{LDR}_i$. Finally, the biochemist consolidates all $\mathcal{LDR}_i$ to form the final laboratory diagnostic result $\mathcal{LDR}$ and delivers it to the $\mathcal{PA}$.

**Imageological Examination**: Based on the preliminary consultation advice, the $\mathcal{PA}$ proceeds to the radiology room to undergo a series of necessary examinations according to the imaging test requirements $\mathcal{ITS}$, such as X-rays, CT scans, or MRI. Then, the radiologist provides the corresponding imaging diagnostic results $\mathcal{IDR}_i$ based on the textual reports $\mathcal{IR}_i$ of the medical images. Finally, the radiologist consolidates all $\mathcal{IDR}_i$ to form the final imaging diagnostic result $\mathcal{IDR}$ and delivers it to the $\mathcal{PA}$.

**Final Consultation**: The $\mathcal{PA}$ returns to the consultation room and provides the $\mathcal{CT}$ with the $\mathcal{LDR}$ and $\mathcal{IDR}$. The $\mathcal{CT}$ discusses the newly obtained $\mathcal{LDR}$ and $\mathcal{IDR}$, with each clinician $\mathcal{CL}_i$ sequentially providing the diagnostic basis $\mathcal{B}_i$, differential diagnosis $\mathcal{D}_i$, final diagnosis $\mathcal{F}_i$, principle of treatment $\mathcal{G}_i$, and treatment plan $\mathcal{T}_i$. Finally, the $\mathcal{CL}_{chair}$ consolidates all evaluations, taking into account the suggestions from all team members, and provides the final $\mathcal{B}, \mathcal{D}, \mathcal{F}, \mathcal{G},$ and $\mathcal{T}$.

**Medical Treatment**: The $\mathcal{PA}$ receives corresponding treatment based on $\mathcal{F}$ and $\mathcal{T}$, including treatments conducted in the treatment room and medication dispensed in the pharmacy. Finally, the $\mathcal{PA}$ leaves the hospital after completing the treatment.

### A.4 EVALUATION METHODS FOR CLINICALAGENT

**Automatic Evaluation** We continue to use the automatic metrics mentioned in Section 4.2 to evaluate the performance of ClinicalAgent, including DWR, CDR, Acceptability, and Avg.

**Human Evaluation** To evaluate the quality and accuracy of the model's diagnostic results, we hire seven medical experts from different departments, with an average of over ten years of clinical experience, to conduct a human evaluation experiments. We randomly select 20 data cases and

**Algorithm 1** ClinicalAgent Algorithm

1: **Input**: Chief Complaint $\mathcal{CC}$, Medical History $\mathcal{MH}$, physical examination $\mathcal{PE}$, Priori Knowledge $\mathcal{PK}$

2: **Output**: Department of the visit $\mathcal{DP}$, Preliminary Diagnosis $\mathcal{P}$, Diagnostic Basis $\mathcal{B}$, Differential Diagnosis $\mathcal{D}$, Final Diagnosis $\mathcal{F}$, Principle of Treatment $\mathcal{G}$, Treatment Plan $\mathcal{T}$

3:

4: % Entry: Hospital

5: $\mathcal{CT} \leftarrow \emptyset$  % Clinician Team

6: $\mathcal{DP}, \mathcal{K}, \mathcal{N} \leftarrow \text{Navigator}(\mathcal{CC})$  % $\mathcal{K}$ is the number of arranged departments, $\mathcal{N}$ is the number of clinicians assigned to each department

7: **for** $k = 1$ to $\mathcal{K}$ **do**

8:     **for** $n = 1$ to $\mathcal{N}$ **do**

9:         $\mathcal{CL}_n^{\mathcal{DP}_k} \leftarrow \text{Mapping}(\mathcal{DP}_k, \mathcal{PK}, n)$     % Retrieve and call clinicians

10:         $\mathcal{CT}.\text{append}(\mathcal{CL}_n^{\mathcal{DP}_k})$

11:     **end for**

12: **end for**

13:

14: % Entry: Consultation Room

15: $\mathcal{CL}_{chair} \leftarrow \text{SelectLeadClinician}(\mathcal{CT})$

16: $\mathcal{P}, \mathcal{LTS}, \mathcal{ITS} \leftarrow \emptyset, \emptyset, \emptyset$  % $\mathcal{P}$ is a set of possible diseases, $\mathcal{LTS}$ is a set of potential laboratory tests that may be required, $\mathcal{ITS}$ is a set of potential imageological tests that may be required

17: **for** $\mathcal{CL}_i$ in $\mathcal{CT}$ **do**

18:     $\mathcal{P}_i, \mathcal{LTS}_i, \mathcal{ITS}_i \leftarrow \mathcal{CL}_i(\mathcal{CC}, \mathcal{MH}, \mathcal{PE})$

19:     $\mathcal{P}.\text{append}(\mathcal{P}_i);$  $\mathcal{LTS}.\text{append}(\mathcal{LTS}_i);$  $\mathcal{ITS}.\text{append}(\mathcal{ITS}_i)$

20: **end for**

21: $\mathcal{P}, \mathcal{LTS}, \mathcal{ITS} \leftarrow \text{Aggregate}(\mathcal{P}, \mathcal{CL}_{chair}), \text{Aggregate}(\mathcal{LTS}, \mathcal{CL}_{chair}), \text{Aggregate}(\mathcal{ITS}, \mathcal{CL}_{chair})$

22:

23: % Entry: Biochemical Testing Room

24: $\mathcal{LDR} \leftarrow \emptyset$  % Laboratory diagnostic results

25: **for** $test_j$ in $\mathcal{LTS}$ **do**

26:     $\mathcal{LR}_j \leftarrow test_j(\mathcal{PA})$  % $\mathcal{LR}_j$ is a laboratory report generated by medical instruments.

27:     $\mathcal{LDR}_j \leftarrow \text{Biochemist}(\mathcal{LR}_j)$

28:     $\mathcal{LDR}.\text{append}(\mathcal{LDR}_j)$

29: **end for**

30:

31: % Entry: Radiology Room

32: $\mathcal{IDR} \leftarrow \emptyset$  % Imaging diagnostic results

33: **for** $test_m$ in $\mathcal{ITS}$ **do**

34:     $\mathcal{IR}_m \leftarrow test_m(\mathcal{PA})$  % $\mathcal{IR}_m$ is a natural language radiological report of medical images.

35:     $\mathcal{IDR}_m \leftarrow \text{Radiologist}(\mathcal{IDR}_m)$

36:     $\mathcal{IDR}.\text{append}(\mathcal{IDR}_m)$

37: **end for**

38:

39: % Back: Consultation Room

40: $\mathcal{B}, \mathcal{D}, \mathcal{F}, \mathcal{G}, \mathcal{T} \leftarrow \emptyset, \emptyset, \emptyset, \emptyset, \emptyset$

41: **for** $\mathcal{CL}_i$ in $\mathcal{CT}$ **do**

42:     $\mathcal{B}_i \leftarrow \mathcal{CL}_i(\mathcal{CC}, \mathcal{MH}, \mathcal{PE}, \mathcal{P}, \mathcal{LDR}, \mathcal{IDR})$

43:     $\mathcal{D}_i \leftarrow \mathcal{CL}_i(\mathcal{CC}, \mathcal{MH}, \mathcal{PE}, \mathcal{P}, \mathcal{LDR}, \mathcal{IDR}, \mathcal{B}_i)$

44:     $\mathcal{F}_i \leftarrow \mathcal{CL}_i(\mathcal{CC}, \mathcal{MH}, \mathcal{PE}, \mathcal{P}, \mathcal{LDR}, \mathcal{IDR}, \mathcal{B}_i, \mathcal{D}_i)$

45:     $\mathcal{G}_i \leftarrow \mathcal{CL}_i(\mathcal{F}_i)$

46:     $\mathcal{T}_i \leftarrow \mathcal{CL}_i(\mathcal{CC}, \mathcal{MH}, \mathcal{PE}, \mathcal{P}, \mathcal{LDR}, \mathcal{IDR}, \mathcal{B}_i, \mathcal{D}_i, \mathcal{F}_i, \mathcal{G}_i)$

47:     $\mathcal{B}.\text{append}(\mathcal{B}_i); \mathcal{D}.\text{append}(\mathcal{D}_i); \mathcal{F}.\text{append}(\mathcal{F}_i); \mathcal{G}.\text{append}(\mathcal{G}_i); \mathcal{T}.\text{append}(\mathcal{T}_i);$

48: **end for**

49: $\mathcal{B}, \mathcal{D}, \mathcal{F} \leftarrow \text{Aggregate}(\mathcal{B}, \mathcal{CL}_{chair}), \text{Aggregate}(\mathcal{D}, \mathcal{CL}_{chair}), \text{Aggregate}(\mathcal{F}, \mathcal{CL}_{chair}),$

50: $\mathcal{G}, \mathcal{T} \leftarrow \text{Aggregate}(\mathcal{G}, \mathcal{CL}_{chair}), \text{Aggregate}(\mathcal{T}, \mathcal{CL}_{chair})$

51:

52: % Entry: Treatment Room

53: $\text{Nurse}(\mathcal{PA}, \mathcal{F}, \mathcal{T})$    % Administer treatment

54:

55: % Entry: Drug Room

56: $\text{Pharmacist}(\mathcal{PA}, \mathcal{F}, \mathcal{T})$    % Dispense medication

provide the experts with diagnostic results from seven different scenarios: ClinicalAgent, two baseline variants, and the four LLMs that perform best in the automatic evaluation. Each diagnostic result is anonymized to ensure that the evaluators cannot identify the corresponding model. Additionally, each sample is graded by two different experts in a double-blind cross-evaluation setting. Following the setup of previous work (Wang et al., 2023), we ask the evaluators to rate the diagnostic results on a scale of 1-5 in four dimensions: fluency, relevance, completeness, and proficiency in medicine. Detailed rubics are provided in Figure 7.

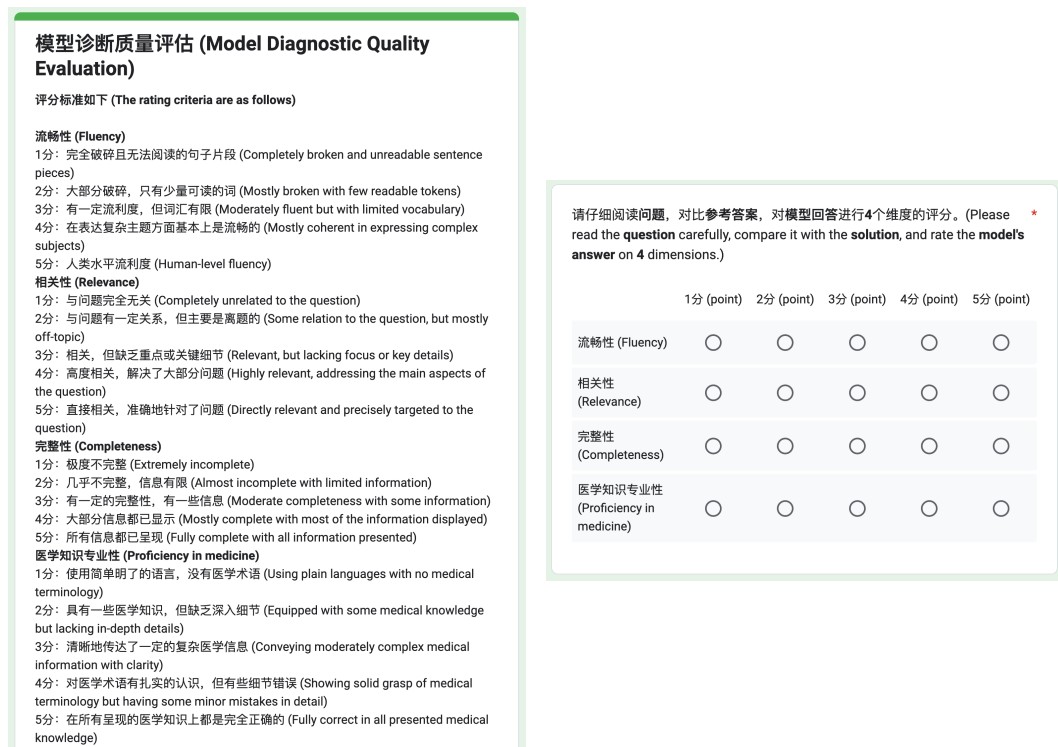

Figure 7: A survey questionnaire for human evaluation of model diagnostic quality, including rating criteria for four dimensions: fluency, relevance, completeness, and proficiency in medical knowledge.

**GPT-4o Evaluation** We randomly select one data sample for each disease, totaling 150 cases. We use GPT-4o to score the diagnostic results from seven models mentioned in Section A.4, following the same rubics as human evaluation.

## A.5 PATIENT NAVIGATOR SETTINGS

Following the experimental design in Section 5.2, we use the Acc@K metric to evaluate navigation accuracy, with K values set at 1, 3, and 5. Additionally, following the description in Section 4.2, we calculate the DIFR-Q and DIFR-N to evaluate instruction following during navigation.

InternLM2, Gemini-Pro, and GPT-4 exhibit outstanding accuracy. Although most models are nearly perfect in the DIFR-Q, they perform poorly in the DIFR-N, especially GPT-4, having a DIFR-N of only 74.03%. Overall, Gemini-Pro and Yi sexhibit remarkable performance. However, Gemini-Pro still has deficiencies in the correctness of department names. We expect the patient navigator to not only accurately guide patients to the appropriate departments but also to minimize errors in department names. Therefore, we choose Yi-Chat as the patient navigator for ClinicalAgent.

## A.6 LIMITATIONS

To date, no medical agents have been evaluated on real clinical data, including the recent MedAgents (Tang et al., 2023) and Agent Hospital (Li et al., 2024). Existing agents are primarily designed

for handling multiple-choice questions and are not suitable for simulating complex real-world scenarios. Specifically, after carefully reviewing the original paper and code of MedAgents, we find that evaluating it would require significant modifications to its experimental design and prompts. These changes would force MedAgents' core process, which is designed for multiple-choice questions, to shift toward a generative question-answering approach, likely impacting its performance negatively. In addition to the previously mentioned differences in design goals, data incompatibility, and variations in evaluation metrics and scope, we are particularly concerned that such comparisons may lead to an unfair evaluation of MedAgents. Therefore, ClinicalAgent cannot be directly compared with other agents. However, as a foundational approach, ClinicalAgent can integrate well with other agents.

We have not conducted practical engineering tests on widely recognized methods such as few-shot learning and RAG, although these methods have proven effective in other medical agents. Future research can consider combining these mature techniques with our ClinicalAgent approach to explore their potential applications in clinical settings. We view this as a direction for future work.

The data in ClinicalBench comes from mainland China and only follows the officially recommended diagnostic methods and procedures in mainland China (NHC, 2022). Therefore, there may be a lack of representativeness for other regions and countries.

## A.7 ETHICAL STATEMENT

Given that our research involves real patient records and their associated diagnostic processes, ethical considerations are central to this study. To mitigate potential ethical and privacy risks, we take the following measures: First, we implement privacy protection measures for all medical records by removing patient identities, treatment area information, and other sensitive content from the data samples. Second, we conduct three rounds of independent and rigorous review of the dataset to ensure it does not contain any sensitive information. Additionally, our study has been approved by the medical ethics committee of the hospital providing the data samples and has been officially notarized by a notary institution. Finally, we require all users of the dataset to agree to the **ClinicalBench Usage and Data Distribution License Agreement** before downloading, which includes a commitment to protect patient privacy and explicit provisions prohibiting the tracking or identifying of any personal information. Through these measures, we ensure the ethical and legal compliance of our research.

We can provide supporting documents and certification from medical institutions and notary organizations to demonstrate the legality and ethical approval of our data collection process. Since the review process is double-blind, we do not include these documents in the supplementary materials. If you need to review them, please contact us to obtain the relevant materials.

---

**Data Example (Chinese)**

【病例编号】: 2e4ff11eaa244c2d8124e537d2b061e3

【科室】: 消化内科

【病例摘要】:
患者基本信息: 中年男性，XX岁。(我们对展示样例的年龄信息进行匿名化处理。)
主诉: 进食后呕血2天。
病史: 患者于2天前进食较硬食物后出现呕血，为咖啡色样胃内容物，共100ml，伴头晕、心慌、乏力，无腹胀、腹痛，无黑便、血便，无意识模糊，就诊给予抑酸、止血保守治疗后呕血症状好转。既往慢性乙型病毒性肝炎3年，未予以治疗。
体格检查: 全身皮肤黏膜苍白，腹部平坦，无胃肠蠕动波，腹式呼吸存在，未见腹壁静脉曲张。腹柔软，无液波震颤，无振水声，腹部包块未触及，无明显压痛和反跳痛，肝脾肋下未触及，Murphy's征阴性，无明显肾区压痛、叩击痛，腹部血管搏动未见明显异常。双侧输尿管压痛点无明显压痛。肝浊音界存在，肝上界位于右锁骨中线第 V 肋间，无移动性浊音。肠鸣音正常。
辅助检查:
  1. 影像学检查:
  1.1. CT平扫+增强: (1) 右肺下叶背段磨玻璃结节，建议3-6个月复查CT。(2) 双肺下叶条索。(3) 肝硬化、脾大，食管下端、胃底静脉曲张，脾门前方静脉曲张。(4) 考虑肝S7被膜下小血管瘤，建议MRI进一步检查。(5) 肝右叶多发小囊肿。(6) 胆囊窝积液。(7) 下腹部CT扫描未见明显异常。
  1.2. 磁共振平扫+增强: (1) 肝硬化、纤维化; 脾大; 门脉高压。(2) 肝S5段小囊肿。(3) 胆囊窝少量积液。
  1.3. 食道、胃、十二指肠镜: (1) 食管静脉曲张破裂出血套扎术及组织胶硬化术。(2) 经胃镜食管药物注射; (3) 经内镜止血术; (4) 门脉高压性胃病。
  2. 实验室检查:
  2.1. 血常规: (1) 红细胞(RBC) $3.0 \times 10^{12}/L \downarrow$; (2) 血红蛋白(HGB) $97g/L \downarrow$; (3) 红细胞比容(HCT) $27.9\% \downarrow$; (4) 血小板计数(阻抗法)(PLT-I) $47 \times 10^9/L \downarrow$; (5) 平均血小板容积(MPV) $13.2fL \uparrow$; (6) 血小板比容(PCT) $0.06\% \downarrow$。
  2.2. 血生化: (1) 天冬氨酸氨基转移酶(AST) $60U/L \uparrow$; (2) 总蛋白(TP) $61.6g/L \downarrow$; (3) 白蛋白(ALB) $31.7g/L \downarrow$; (4) 白蛋白/球蛋白比值(A/G) $1.11.2 - 2.4 \downarrow$; (5) 总胆红素(TBIL) $41.5mol/L \uparrow$; (6) 直接胆红素(DBIL) $10.0mol/L \uparrow$; (7) 间接胆红素(IBIL) $31.5mol/L \uparrow$; (8) 前白蛋白(PA) $93.5mg/L \downarrow$; (9) 钙(Ca) $2.10mmol/L \downarrow$; (10) 钠(Na) $136mmol/L \downarrow$; (11) 渗透压(OSM) $272mOsm/kg \downarrow$。
  2.3. 凝血功能检测: (1) 凝血酶原时间#(PT#) $20.8S \uparrow$; (2) 凝血酶时间#(TT#) $19.5S \uparrow$; (3) 纤维蛋白原#(Fg#) $1.1g/L \downarrow$; (4) 百分活度(PT%) $43\% \downarrow$; (5) 国际标准化比值(PT.INR) $1.810.85 - 1.25 \uparrow$。
  2.4. 肿瘤标志物检测: (1) 甲胎蛋白(AFP) $307.2ng/mL \uparrow$; (2) 糖链抗原19-9(CA19-9) $69.9U/mL \uparrow$。

【临床诊断】:
  1. 初步诊断: (1) 上消化道出血; (2) 食管胃底静脉曲张破裂出血; (3) 肝硬化; (4) 贫血; (5) 电解质紊乱。
  2. 诊断依据:
  2.1. 患者慢性乙型病毒性肝炎病史，进食硬质食物后呕血2天。
  2.2. 体格检查支持诊断: (1) 腹部平坦，无胃肠蠕动波，腹式呼吸存在，未见腹壁静脉曲张; (2) 腹柔软，无液波震颤，无振水声，腹部包块未触及，无明显压痛和反跳痛，肝脾肋下未触及，Murphy's征阴性，无明显肾区压痛、叩击痛，腹部血管搏动未见明显异常; (3) 双侧输尿管压痛点无明显压痛。肝浊音界存在，肝上界位于右锁骨中线第 V 肋间，无移动性浊音; (4) 肠鸣音正常。
  2.3. 影像学检查支持诊断: (1) CT平扫+增强示: 肝硬化、脾大，食管下端、胃底静脉曲张，脾门前方静脉曲张; (2) 磁共振平扫+增强示: 肝硬化、纤维化; 脾大; 门脉高压; (3) 食道、胃、十二指肠镜示: 食管静脉曲张破裂出血; 门脉高压性胃病。
  2.4. 实验室检查支持诊断: (1) 血常规示: 红细胞(RBC)降低，血红蛋白(HGB)降低，红细胞比容(HCT)降低; (2) 血生化示: 天冬氨酸氨基转移酶(AST)升高，总蛋白(TP)降低，白蛋白(ALB)降低，白蛋白/球蛋白比值(A/G)降低，总胆红素(TBIL)升高，直接胆红素(DBIL)升高，间接胆红素(IBIL)升高，前白蛋白(PA)降低，钙(Ca)降低，钠(Na)降低，渗透压(OSM)降低; (3) 凝血功能检测示: 凝血酶原时间#(PT#)升高，凝血酶时间#(TT#)升高，纤维蛋白原#(Fg#)降低，百分活度(PT%)降低，国际标准化比值(PT.INR)升高。

Table 5: A data example from ClinicalBench.

Data Example (Chinese)

3. 鉴别诊断:

3.1. 胃十二指肠溃疡并出血: 是溃疡病常见并发症, 小量出血往往没有临床症状, 仅在大便隐血试验时发现。出血量大于500ml, 即为大出血, 主要表现为呕血、便血和不同程度的贫血。有溃疡病史者, 发现有胃肠道大出血, 首先应考虑为胃十二指肠溃疡出血。

3.2. 贲门黏膜撕裂: 胃食管交界处或贲门处的纵向粘膜撕裂致上消化道出血, 85%的病人有呕血症状。典型的表现是在恶心或呕吐后出现呕血。胃镜检查可通过在胃食管连接处内或附近的粘膜裂口上发现活动性出血, 粘附的血块或纤维蛋白结皮来进行诊断。

3.3. 消化道肿瘤并出血: 约5%可发生大出血, 表现为呕血和(或)黑便, 常见于40岁以上, 特别是男性, 近期出现消化不良、腹痛等消化道症状, 既往有消化道肿瘤病史或消化道肿瘤家族史的患者。

4. 最终诊断: 食管胃底静脉曲张破裂出血。

5. 治疗原则: (1) 维持呼吸及循环系统功能正常; (2) 控制出血情况, 介入或外科手术治疗。

6. 治疗计划: (1) 根据患者病情予以建立静脉通路、禁食水、监测生命体征。(2) 治疗上给予静脉滴注奥美拉唑及生长抑素止血抑酸护胃; 头孢曲松预防感染、异甘草酸镁改善肝功能异常; 定期监测血常规, 必要时输血治疗; 补液维持水电解质及酸碱平衡稳定以及营养支持等对症处理治疗。(3) 完善心电图、心脏彩超等入院常规检查, 明确手术指征, 排除手术禁忌症后择期行内镜下手术治疗。

【CT平扫+增强】:

1. 影像所见: (1) 肺部: 右肺下叶背段见磨玻璃结节影, 大小约5mm×4mm。双肺下叶可见条索样高密度影。(2) 纵隔: 两侧肺门结构显示尚可, 气管及支气管通畅。纵隔内未见明显肿大淋巴结影。心脏大小、形态、位置未见明显异常。双侧无胸膜增厚。食管下端、胃底可见多发增宽、迂曲血管影。(3) 肝脏: 肝左叶体积增大, 实质密度不均匀, 肝脏边缘欠规整。肝S7被膜下可见小斑片状稍高密度影, 直径约1.1cm, 增强后呈渐进性强化。肝右叶可见多发小类圆形低密度影, 较大者直径约0.7cm, 增强后未见强化。肝内外胆管未见扩张。(4) 胆囊: 大小如常, 壁无增厚, 腔内未见异常密度, 胆囊窝可见积液影。(5) 脾脏: 脾脏体积增大, 未见明显异常强化影, 脾门前方可见多发增宽、迂曲血管影。(6) 胰腺: 轮廓清楚, 形态大小如常, 未见异常密度影及胰管扩张。(7) 肾上腺区: 未见明显异常改变。(8) 双肾: 双肾对称, 形态大小如常, 未见异常密度影。(9) 腹、盆腔: 腹腔及腹膜后间隙未见肿大淋巴结影。前列腺形态大小正常, 其内未见异。常密度影, 双侧精囊腺大小、形态及密度未见异常。膀胱充盈可, 壁无增厚, 腔内未见异。常密度影, 双侧盆壁及腹股沟未见肿大淋巴结。

2. 影像诊断: (1) 右肺下叶背段磨玻璃结节, 建议3-6个月复查CT。(2) 双肺下叶条索。(3) 肝硬化、脾大, 食管下端、胃底静脉曲张, 脾门前方静脉曲张。(4) 考虑肝S7被膜下小血管瘤, 建议MRI进一步检查。(5) 肝右叶多发小囊肿。(6) 胆囊窝积液。(7) 下腹部CT扫描未见明显异常。

【磁共振平扫+增强】:

1. 影像所见: 肝脏体积不大, 内弥散分布细条、网状索状T2压脂高信号影; 肝S5段小圆形T2高信号影, 直径约为6mm。胆囊外形不大, 内未见明显异常信号影; 胆囊窝内少量液性信号影。脾脏体积显著增大, 内信号均匀。胰腺、双肾外形规则, 信号均匀。贲门胃底旁可见扭曲小血管影。门脉、脾静脉增粗。

2. 影像诊断: (1) 肝硬化、纤维化。(2) 脾大。(3) 门脉高压。(4) 肝S5段小囊肿。(5) 胆囊窝少量积液。

【食道、胃、十二指肠镜】:

1. 影像所见: 食道通过顺利, 下段静脉中度曲张, 呈串珠样, 红色征阳性。食管曲张静脉套扎机套扎5环。贲门开合佳通畅, 食管静脉曲张延续至胃底, 可见团状静脉曲张。三明治法, 分两点注入一点注入聚桂醇10ml, 组织胶3ml(6支); 另一点注入聚桂醇10ml, 组织胶3ml(6支); 胃体粘膜红肿糜烂。胃窦粘膜充血水肿, 红白相间, 以红为主, 可见散在小片状糜烂, 幽门圆, 开放好, 十二指肠球部未见明显异常。

2. 影像诊断: (1) 食管静脉曲张破裂出血套扎术及组织胶硬化术。(2) 经胃镜食管药物注射。(3) 经内镜止血术。(4) 门脉高压性胃病。

Table 6: A data example from ClinicalBench. (Cont. Table 5)

---

### Data Example (English)

[Case ID]: 2e4ff11eaa244c2d8124e537d2b061e3

[Department]: Gastroenterology

[Case Summary]:
Patient Basic Information: Middle-aged male, XX years old. (We anonymize the age information in the sample data presented.)
Chief Complaint: Vomiting blood for 2 days after eating.
Medical History: The patient experienced vomiting of coffee-colored gastric contents (approximately 100ml) accompanied by dizziness, palpitations, and weakness after consuming hard food 2 days ago. There was no abdominal distension, pain, melena, or bloody stool, nor any confusion. The patient was treated conservatively with acid-suppressing and hemostatic medications, after which symptoms of vomiting blood improved. The patient has a history of chronic Hepatitis B for three years, which has not been treated.
Physical Examination: Pale skin and mucous membranes, flat abdomen with no visible peristaltic waves and presence of abdominal breathing. No abdominal wall vein varicosity was observed. The abdomen was soft without fluid wave or shifting dullness, and no palpable masses. There was no significant tenderness or rebound tenderness, and the liver and spleen were not palpable below the ribs. Murphy's sign was negative. No evident kidney area tenderness or percussion pain, and no abnormal vascular pulsation in the abdomen. No significant tenderness at bilateral ureteral pressure points. Liver dullness was present, with the upper boundary at the right mid-clavicular line at the fifth intercostal space, with no shifting dullness. Bowel sounds were normal.
Auxiliary Examination:
    1. Imaging Examination:
    1.1. CT Scan (Plain + Contrast): (1) Ground glass nodule in the lower lobe of the right lung, suggest a follow-up CT in 3-6 months. (2) Linear opacities in the lower lobes of both lungs. (3) Liver cirrhosis, splenomegaly, varices at the lower end of the esophagus and the gastric fundus, and varices in front of the spleen. (4) Possible subcapsular hemangioma in liver segment S7, further examination with MRI suggested. (5) Multiple small cysts in the right lobe of the liver. (6) Fluid accumulation in the gallbladder fossa. (7) No apparent abnormalities in the lower abdominal CT scan.
    1.2. MRI (Plain + Contrast): (1) Liver cirrhosis, fibrosis; enlarged spleen; portal hypertension. (2) Small cyst in liver segment S5. (3) Minor fluid accumulation in the gallbladder fossa.
    1.3. Esophagogastroduodenoscopy: (1) Esophageal varices rupture treated with banding and tissue glue sclerotherapy. (2) Esophageal drug injection via gastroscopy. (3) Endoscopic hemostasis. (4) Portal hypertensive gastropathy.
    2. Laboratory Examination:
    2.1. Routine Blood Test: (1) Red Blood Cells (RBC) $3.0 \times 10^{12}/L$ ↓; (2) Hemoglobin (HGB) $97g/L$ ↓; (3) Hematocrit (HCT) 27.9% ↓; (4) Platelet Count (Impedance method) (PLT-I) $47 \times 10^9/L$ ↓; (5) Mean Platelet Volume (MPV) 13.2fL ↑; (6) Plateletcrit (PCT) 0.06% ↓.
    2.2. Blood Biochemistry Test: (1) Aspartate Aminotransferase (AST) $60U/L$ ↑; (2) Total Protein (TP) $61.6g/L$ ↓; (3) Albumin (ALB) $31.7g/L$ ↓; (4) Albumin/Globulin Ratio (A/G) $1.11.2 - 2.4$ ↓; (5) Total Bilirubin (TBIL) $41.5mol/L$ ↑; (6) Direct Bilirubin (DBIL) $10.0mol/L$ ↑; (7) Indirect Bilirubin (IBIL) $31.5mol/L$ ↑; (8) Prealbumin (PA) $93.5mg/L$ ↓; (9) Calcium (Ca) $2.10mmol/L$ ↓; (10) Natrium (Na) $136mmol/L$ ↓; (11) Osmolality (OSM) $272mOsm/kg$ ↓.
    2.3. Coagulation Function Test: (1) Prothrombin Time# (PT#) $20.8S$ ↑; (2) Thrombin Time# (TT#) $19.5S$ ↑; (3) Fibrinogen# (Fg#) $1.1g/L$ ↓; (4) Prothrombin Activity (PT%) 43% ↓; (5) International Normalized Ratio (PT.INR) $1.810.85 - 1.25$ ↑.
    2.4. Tumor Marker Test: (1) Alpha-Fetoprotein (AFP) $307.2ng/mL$ ↑; (2) Carbohydrate Antigen 19-9 (CA19-9) $69.9U/mL$ ↑.

[Clinical Diagnosis]:
    1. Preliminary Diagnosis: (1) Upper gastrointestinal bleeding; (2) Uptured esophagogastric varices bleeding; (3) Liver cirrhosis; (4) Anemia; (5) Electrolyte imbalance.

Table 7: A data example from ClinicalBench. (Cont. Table 6)

> ### Data Example (English)
>
> 2. Diagnostic Basis:
>
> 2.1. History of chronic Hepatitis B, and vomiting blood for 2 days after eating hard food.
>
> 2.2. Physical examination supports the diagnosis: (1) Flat abdomen, no gastrointestinal peristaltic waves, abdominal breathing present, no visible abdominal wall vein varicosity. (2) Soft abdomen, no fluid wave or shifting dullness, no palpable masses, no significant tenderness or rebound pain, liver and spleen not palpable below the ribs, Murphy's sign negative, no evident renal tenderness or percussion pain, no abnormal vascular pulsation in the abdomen. (3) No significant tenderness at bilateral ureteral pressure points. Liver dullness present, upper boundary at the right mid-clavicular line at the fifth intercostal space, no shifting dullness. (4) Normal bowel sounds.
>
> 2.3. Imaging examinations support the diagnosis: (1) CT Scan (Plain + Contrast) showing liver cirrhosis, splenomegaly, varices at the lower end of the esophagus and the gastric fundus, and varices in front of the spleen. (2) MRI (Plain + Contrast) indicating liver cirrhosis, fibrosis, enlarged spleen, portal hypertension. (3) Endoscopy (Esophagus, Stomach, Duodenum) revealing ruptured esophageal varices and portal hypertensive gastropathy.
>
> 2.4. Laboratory examinations support the diagnosis: (1) Routine Blood Test shows: decreased red blood cells (RBC), decreased hemoglobin (HGB), and decreased hematocrit (HCT). (2) Blood biochemistry Test shows: increased aspartate aminotransferase (AST), decreased total protein (TP), decreased albumin (ALB), decreased albumin/globulin ratio (A/G), increased total bilirubin (TBIL), increased direct bilirubin (DBIL), increased indirect bilirubin (IBIL), decreased prealbumin (PA), decreased calcium (Ca), decreased natrium (Na), and decreased osmolarity (OSM). (3) Coagulation Function Test shows: increased prothrombin time (PT), increased thrombin time# (TT#), decreased fibrinogen# (Fg#), decreased percent activity (PT%), and increased International Normalized Ratio (PT.INR).
>
> 3. Differential Diagnosis:
>
> 3.1. Gastric and Duodenal Ulcer with Bleeding: Bleeding is a common complication of ulcer disease. Minor bleeding often presents with no clinical symptoms and is only detected during fecal occult blood tests. A bleed greater than 500ml is considered severe, primarily manifested as vomiting blood, bloody stools, and varying degrees of anemia. In patients with a history of ulcer disease presenting with significant gastrointestinal bleeding, gastric and duodenal ulcers should be the first consideration.
>
> 3.2. Mallory-Weiss Tear: This condition involves a longitudinal mucosal tear at the gastroesophageal junction or cardia leading to upper gastrointestinal bleeding, with 85% of patients presenting with symptoms of vomiting blood. The typical presentation occurs after an episode of nausea or vomiting. Gastroscopy can diagnose this condition by identifying active bleeding, adherent blood clots, or a fibrin crust on or near the mucosal tear at the gastroesophageal junction.
>
> 3.3. Gastrointestinal Tumor with Bleeding: About 5% of cases may experience significant bleeding, presenting as vomiting blood and/or melena (black stools). It is commonly seen in individuals over 40 years old, especially males, who have recently experienced poor general condition, abdominal pain, or other gastrointestinal symptoms. Patients with a personal or family history of gastrointestinal tumors should be particularly considered.
>
> 4. Final Diagnosis: Rupture and bleeding of esophagogastric varices.
>
> 5. Principle of Treatment: (1) Maintain normal respiratory and circulatory system function. (2) Control bleeding conditions, interventional or surgical treatment.
>
> 6. Treatment Plan: (1) Based on the patient's condition, establish intravenous access, withhold food and water, and monitor vital signs. (2) For treatment, administer intravenous infusion of omeprazole and somatostatin to stop bleeding and protect the stomach from acid; ceftriaxone to prevent infection, and magnesium isoglycyrrhizinate to improve liver function abnormalities; regularly monitor complete blood count, and perform blood transfusion treatment when necessary; provide fluid replacement to maintain stability of electrolytes and acid-base balance, as well as nutritional support and other symptomatic treatments. (3) Complete routine admission tests such as electrocardiograms and cardiac echocardiography, determine surgical indications, rule out contraindications for surgery, and then schedule endoscopic surgery when appropriate.

Table 8: A data example from ClinicalBench. (Cont. Table 7)

---

**Data Example (English)**

[CT Scan (Plain + Contrast)]:

1. Findings: (1) Lungs: There is a ground-glass nodule in the dorsal segment of the right lower lobe, approximately 5mm x 4mm in size. There are strip-like high-density shadows in both lower lobes. (2) Mediastinum: The structures of both hilum are normal; trachea and bronchi are patent. No significantly enlarged lymph nodes seen in the mediastinum. The heart is normal in size, shape, and position. No pleural thickening on both sides. Dilated and tortuous vessels are visible at the lower end of the esophagus and the fundus of the stomach. (3) Liver: Increased volume of the left hepatic lobe with uneven parenchymal density and irregular liver margins. A small patchy slightly hyperdense shadow is seen subcapsularly in liver segment S7, about 1.1cm in diameter, showing progressive enhancement post-contrast. Multiple small round hypo-dense shadows are seen in the right lobe, the largest being about 0.7cm in diameter, with no enhancement post-contrast. No dilatation of the intrahepatic and extrahepatic bile ducts. (4) Gallbladder: Normal size, no wall thickening, no abnormal density within the lumen, fluid seen in the gallbladder fossa. (5) Spleen: Enlarged spleen with no obvious abnormal enhancement, multiple dilated and tortuous vascular shadows anterior to the hilum. (6) Pancreas: Clear outline, normal shape and size, no abnormal density or pancreatic duct dilation. (6) Adrenal Area: No significant abnormalities. (7) Kidneys: Symmetrical kidneys, normal in shape and size, no abnormal density. (8) Abdomen and Pelvis: No enlarged lymph nodes in the abdominal cavity and retroperitoneal space. Normal prostate morphology and size, no abnormalities within. Normal seminal vesicle glands in size, shape, and density. The bladder is well-filled, with no wall thickening, and no abnormal density within. No enlarged lymph nodes in both pelvic walls and inguinal areas.

2. Impression: (1) Ground-glass nodule in the dorsal segment of the right lower lobe, recommend follow-up CT in 3-6 months. (2) Strip-like densities in both lower lobes. (3) Cirrhosis, splenomegaly, esophageal and gastric fundal varices, varices anterior to the spleen hilum. (4) Possible small hemangioma subcapsularly in liver segment S7, recommend further examination with MRI. (5) Multiple small cysts in the right lobe of the liver. (6) Fluid in the gallbladder fossa. (7) No significant abnormalities in the lower abdominal CT scan.

[MRI Scan (Plain + Contrast))]:

1. Findings: (1) Liver: Not large in volume, with diffuse distribution of thin, reticular high signal T2 fat-suppressed strands; small round high signal T2 lesion in liver segment S5, about 6mm in diameter. Gallbladder is small, with no significant abnormal signal within; a small amount of liquid signal in the gallbladder fossa. (2) Spleen: Significantly enlarged, with uniform signal. (3) Pancreas and Kidneys: Regular shape, uniform signal. (4) Adjacent to the gastroesophageal junction and gastric fundus: Twisted small vascular shadows. Portal vein and splenic vein are thickened.

2. Impression: (1) Cirrhosis, fibrosis. (2) Splenomegaly. (3) Portal hypertension.(4) Small cyst in liver segment S5. (5) Small amount of fluid in the gallbladder fossa.

[Esophagogastroduodenoscopy]:

1. Findings: The passage through the esophagus was smooth, with moderate varices in the lower segment appearing beaded and exhibiting positive red signs. Five rings of esophageal variceal ligation were performed using a variceal banding device. The gastroesophageal junction was well-functioning and patent, with esophageal varices extending to the fundus of the stomach, where cluster-like varices were visible. Sandwich method applied: two sites injected with 10 ml of polidocanol each and 3 ml of tissue adhesive (6 vials each). The gastric body mucosa was inflamed and eroded. The mucosa of the gastric antrum was congested and edematous, primarily red with interspersed white, showing scattered small patches of erosion. The pylorus was round and well-functioning; no obvious abnormalities were observed in the duodenal bulb.

2. Impression: (1) Esophageal variceal rupture with banding and tissue adhesive sclerotherapy. (2) Esophagogastroscopic medication injection. (3) Endoscopic hemostasis. (4) Portal hypertensive gastropathy.

Table 9: A data example from ClinicalBench. (Cont. Table 8)

> ## Case Study (Chinese)
>
> 【病例编号】: 94d8abab8a4643cc91c2443e96f00027
>
> 【科室】: 肝胆胰外科
>
> 【病例摘要】:
> 患者基本信息: 中年男性, XX岁。(我们对展示样例的年龄信息进行匿名化处理。)
> 主诉: 左上腹痛2小时。
> 病史: 患者于2小时前驾车时被方向盘撞击左上腹后出现左上腹疼痛, 为持续剧烈疼痛, 无加重或缓解因素, 不伴头痛不适、意识不清, 无恶心呕吐, 胸闷气促、发热乏力, 尿频尿急等伴随症状, 双侧肋骨骨折, 由120送至我院急诊, 查腹部彩超提示"脾脏异常回声, 考虑血肿", 为求进一步诊治, 以"创伤性脾破裂"收入我科。患者车祸外伤史明确。既往体健。
> 体格检查: 神志清楚, 表情痛苦, 查体欠合作。双肺呼吸音粗, 可闻及湿性音, 心律齐, 未闻及病理性杂音, 腹部平坦, 无胃肠蠕动波, 腹式呼吸存在, 未见腹壁静脉曲张。腹肌紧张, 左上腹有明显压痛, 反跳痛及肌紧张, 肝浊音界存在, 肝上界位于右锁骨中线第V肋间, 肠鸣音减弱, 余腹部查体因疼痛拒查。四肢肌力V级, 肌张力正常, 双下肢无水肿。
> 辅助检查:
>   1. 影像学检查:
>   1.1. 彩色多普勒超声: (1) 脾脏、左肾间异常回声, 局部包裹, 考虑血肿。(2) 脂肪肝。(3) 脾脏回声欠均匀, 提示挫裂伤可能。(4) 腹腔可显示部分未见明显积液; 请结合临床, 必要时常规超声复查。
>   2. 实验室检查:
>   2.1. 血常规: (1) 白细胞(WBC)$10.8 \times 10^9/L$↑; (2) 淋巴细胞百分数(LYMPH%) 11.9%↓; (3) 中性粒细胞百分数(NEUT%) 78.7%↑; (4) 单核细胞绝对值(MONO#) $0.64 \times 10^9/L$↑; (5) 中性粒细胞绝对值(NEUT#) $8.5 \times 10^9/L$↑; (6) 红细胞(RBC) $3.4 \times 10^{12}/L$↓; (7) 血红蛋白(HGB) $104g/L$↓; (8) 红细胞比容(HCT) 31.0%↓; (9) 平均血小板容积(MPV) $10.1fL$↑; (10) 嗜碱细胞绝对值(BASO#) $0.07 \times 10^9/L$↑; (11) C反应蛋白(CRP) $45.77mg/L$↑。
>   2.2. 血生化: (1) 总蛋白(TP) $55.4g/L$↓; (2) 白蛋白(ALB) $35.1g/L$↓; (3) 钙(Ca) $2.06mmol/L$↓。
>
> 【彩色多普勒超声】:
>   影像所见: (1) 肝脏: 形态大小正常, 被膜光滑连续, 实质回声细密、增强, 肝内管状结构走行清晰自然, 门静脉主干内径正常。(2) 胆囊: 形态大小正常, 壁尚光滑, 囊腔内未见异常回声, 肝内外胆管无扩张。(3) 脾脏: 形态大小正常, 实质回声不均匀。脾脏、左肾间可见一片状不规则低回声区, 边界欠清, 内部回声不均, 范围约7.9×4.8cm。(4) 胰腺: 形态大小正常, 实质回声均匀, 主胰管无扩张。(5) 肾脏: 双肾位置、形态、大小正常, 轮廓光滑规整, 皮髓质界限清晰, 实质回声分布均匀, 集合系无分离。(6) CDFI: 血流信号未见明显异常。平卧位探查腹腔, 腹腔可显示部分未见明显游离液性暗区。
>
> 【医生提供的临床诊断和影像诊断】:
>   1. 初步诊断: (1) 创伤性脾破裂; (2) 肋骨骨折; (3) 创伤性肺炎。
>   2. 诊断依据:
>   2.1. 患者外伤后左上腹痛2小时。
>   2.2. 体格检查支持诊断: 体格检查支持诊断。神志清楚, 表情痛苦, 查体欠合作。双肺呼吸音粗, 可闻及湿性音, 心律齐, 未闻及病理性杂音, 腹部平坦, 无胃肠蠕动波, 腹式呼吸存在, 未见腹壁静脉曲张。腹肌紧张, 左上腹有明显压痛, 无明显反跳痛及肌紧张, 肝浊音界存在, 肝上界位于右锁骨中线第V肋间, 肠鸣音减弱, 余腹部查体因疼痛拒查。四肢肌力V级, 肌张力正常, 双下肢无水肿。
>   2.3. 影像学检查支持诊断: (1) 彩色多普勒超声示: 脾脏、左肾间异常回声, 局部包裹, 考虑血肿, 较前范围变化不大; 脂肪肝; 脾脏回声欠均匀, 提示挫裂伤可能; 腹腔可显示部分未见明显积液; 请结合临床, 必要时常规超声复查。
>   2.4. 实验室检查支持诊断: (1) 血常规示: 白细胞(WBC)升高, 淋巴细胞百分数(LYMPH%)降低, 中性粒细胞百分数(NEUT%)升高, 单核细胞绝对值(MONO#)升高, 中性粒细胞绝对值(NEUT#)升高, 红细胞(RBC)降低, 血红蛋白(HGB)降低, 红细胞比容(HCT)降低, 平均血小板容积(MPV)升高, 嗜碱细胞绝对值(BASO#)升高, C反应蛋白(CRP)升高; (2) 血生化示: 总蛋白(TP)降低, 白蛋白(ALB)降低, 钙(Ca)降低。

Table 10: A detailed case study on ClinicalBench.

---

**Case Study (Chinese)**

3. 鉴别诊断:

3.1. 消化性溃疡急性穿孔: 有较典型的溃疡病史, 腹痛突然加剧, 腹肌紧张, 肝浊音界消失, X线透视见膈下有游离气体等可资鉴别。

3.2. 胆石症和急性胆囊炎: 常有胆绞痛史, 疼痛位于右上腹, 常放射到右肩部, Murphy征阳性, 血及尿淀粉酶轻度升高, B超及X线胆道造影可明确诊断。

3.3. 急性坏死性肠炎: 具有腹痛、腹胀、便血、高热及呕吐等症状, 腹胀较严重, 全身中毒症状重。

4. 最终诊断: 脾破裂。

5. 治疗原则: (1) 先保命后保脾; (2) 年龄越小越优先保脾; (3) 根据脾脏损伤程度选择一种或几种保脾方法; (4) 施行脾保留手术后应注意严密观察, 防止出现延迟性脾破裂; (5) 对高龄、一般状态差、严重多发伤、凝血酶原时间显著延长者, 建议施行脾切除术。

6. 治疗计划: (1) 根据患者病情予以特级护理, 心电监护, 建立静脉通路, 报病危, 密切监测生命体征变化。(2) 治予以补液补充血容量。(3) 明确手术指征, 排除手术禁忌症后急诊行手术治疗。

7. 影像诊断: (1) 脾脏、左肾间异常回声, 局部包裹, 考虑血肿, 较前范围变化不大。(2) 脂肪肝。(3) 脾脏回声欠均匀, 提示挫裂伤可能。(4) 腹腔可显示部分未见明显积液; 请结合临床, 必要时常规超声复查。

【Agent#1@3提供的科室导诊、临床诊断和影像诊断】:

1. 科室导诊: 肝胆胰外科。

2. 初步诊断: (1) 外伤性脾破裂伴血肿形成; (2) 腹部闭合性损伤; (3) 双侧肋骨骨折; (4) 轻度贫血; (5) 轻度低钙血症; (6) 脂肪肝。

3. 诊断依据: (1) 患者有明确的车祸外伤史, 特别是方向盘撞击导致的左上腹剧烈疼痛和左侧肋骨骨折。(2) 出现剧烈腹痛, 腹肌紧张、压痛及反跳痛, 提示可能存在腹内脏器损伤。(3) 彩色多普勒超声提示脾脏回声欠均匀, 考虑挫裂伤可能, 以及脾脏、左肾间的异常回声, 局部包裹, 考虑为血肿。(4) 实验室检查显示白细胞总数升高, 中性粒细胞比例增加, C反应蛋白显著升高, 提示存在急性炎症反应; 血红蛋白和红细胞计数降低, 提示轻度贫血; 血钙降低, 提示轻度低钙血症。(5) 超声未见明显腹腔积液, 但不能完全排除肝挫伤和其他内脏损伤。

4. 鉴别诊断:

4.1. 胰腺损伤: 虽然超声未见明显腹腔积液, 但胰腺损伤需进一步排查, 因其症状与脾破裂相似, 可通过CT等影像学检查予以排除。

4.2. 肠破裂或穿孔: 腹膜炎体征需与小肠或结肠穿孔相鉴别, 通过CT或立位腹平片检查可帮助诊断。

4.3. 肾脏损伤: 超声提示脾脏、左肾间异常回声, 需进一步检查明确肾脏是否有损伤。

4.4. ==血小板减少症: 血小板计数虽在正常范围内, 但MPV升高, 提示可能存在血小板功能异常而非数量减少。== 【人类医生评价】: 腹腔内出血导致贫血, 无需与血液疾病鉴别。

5. 最终诊断: 外伤性脾破裂伴血肿形成。

6. 治疗原则: (1) 紧急处理: 评估患者生命体征, 维持气道通畅, 纠正休克, 控制出血。(2) ==手术治疗: 对于脾破裂伴血肿形成, 应尽快进行手术治疗, 评估脾脏损伤情况, 决定是否行脾切除或脾修补术。== 【人类医生评价】: 脾破裂如不严重可绝对卧床保守观察。(3) 抗感染治疗: 预防和控制感染, 给予广谱抗生素治疗。(4) 纠正贫血和电解质紊乱: 输血、补充血制品以及纠正低钙血症。(5) 监测生命体征和器官功能: 密切观察患者的生命体征、尿量、心电图、血气分析等, 及时发现并处理可能出现的并发症。

7. 治疗计划: (1) 完善腹部CT检查, 明确脾脏损伤程度、是否存在其他脏器损伤 (如肝挫伤) 以及肾脏状况。(2) 如证实脾破裂严重, 立即安排急诊剖腹探查术, 术中根据脾脏损伤情况决定是否行脾切除或脾修补术, 并处理其他可能的脏器损伤。(3) 在术前、术后给予广谱抗生素预防感染。(4) 根据血红蛋白和血细胞比容的变化, 适时输注红细胞悬液, 补充血容量。(5) 给予补钙治疗, 纠正低钙血症。(6) 术后严密监测患者生命体征、腹部体征及实验室指标变化, 调整治疗方案。==如果肝挫伤严重, 也需要相应处理。== 【人类医生评价】: 无肝挫伤的诊断。

8. 影像诊断: (1) 肝脏、胆囊、胰腺及双肾的形态、大小、结构、血流信号以及肝内外胆管、门静脉主干、集合系统等方面均显示正常。(2) 然而, 脾脏的实质回声呈现不均匀性改变, 同时在脾脏与左肾之间发现了一片状不规则低回声区域, 边界欠清晰, 内部回声也不均匀, 大小约为7.9×4.8cm。这个发现提示可能存在脾脏局部的占位性病变, 如结节、肿瘤或其他炎症性病灶。为了明确诊断, 建议进一步进行CT、MRI等相关检查或者组织活检。(3) 此外, 腹腔内未见明显游离液性暗区, 排除了大量腹水的可能性。综上所述, 影像学表现为脾脏局部异常及脾肾间隙不规则低回声区, 需进一步排查脾脏占位性病变。其他相关器官未见明显异常。

Table 11: A detailed case study on ClinicalBench, diagnosed by Agent#1@3.

> **Case Study (Chinese)**
>
> 【Agent#3@1提供的科室导诊、临床诊断和影像诊断】:
>  1. 科室导诊: 肝胆胰外科。
>  2. 初步诊断: (1) 外伤性脾破裂伴血肿形成; (2) 左肾挫裂伤; 【人类医生评价】: 彩超未提示肾损伤。 (3) 脂肪肝; (4) 双侧肋骨骨折; (5) 轻度贫血; (6) 轻度低钙血症。
>  3. 诊断依据: (1) 患者有明确的车祸外伤史, 特别是撞击导致的方向盘失控和双侧肋骨骨折。(2) 出现剧烈腹痛, 腹肌紧张、压痛及反跳痛, 提示可能存在腹内脏器损伤。(3) 彩色多普勒超声提示脾脏回声欠均匀, 考虑挫裂伤可能, 脾脏、左肾间异常回声, 局部包裹, 考虑血肿形成。(4) 实验室检查显示白细胞总数升高, 中性粒细胞比例增加, C反应蛋白显著升高, 提示存在急性炎症反应; 血红蛋白和红细胞计数降低, 提示轻度贫血; 血钙降低。(5) 超声未见明显腹腔积液, 但未明确提及肝脏挫裂伤, 肝浊音界存在, 提示肝脏可能未受重伤。
>  4. 鉴别诊断:
>  4.1. 胰腺损伤: 虽超声未见明显腹腔积液, 但胰腺损伤需进一步排查, 可通过CT等影像学检查予以排除。
>  4.2. 肝脏挫裂伤: 超声提示脂肪肝, 但未明确提及肝脏挫裂伤, 需结合临床表现和进一步影像学检查以排除。
>  4.3. 肠破裂: 患者无明显腹膜炎症状, 且超声未提示肠管异常, 故暂不考虑肠破裂。
>  4.4. 胆囊损伤: 患者无典型的胆囊损伤症状, 如黄疸、右上腹痛等, 暂不考虑。
>  4.5. 肾挫裂伤: 超声提示左肾间异常回声 【人类医生评价】: 脾与肾之前的异常回声, 而不是肾异常回声。, 考虑血肿形成, 因此已纳入初步诊断。
>  5. 最终诊断: 外伤性脾破裂伴血肿形成。
>  6. 治疗原则: (1) 紧急处理: 维持生命体征稳定, 包括吸氧、心电监护、建立静脉通路、输液、输血等支持治疗。(2) 手术治疗: 根据病情评估, 可能需要进行剖腹探查术以明确损伤情况, 并进行脾切除或脾修补术, 同时评估并处理其他脏器损伤。(3) 非手术治疗: 对于症状较轻、生命体征稳定的患者, 可以先尝试保守治疗, 包括观察、输液、使用抗生素等。(4) 抗感染治疗: 预防和控制感染, 给予广谱抗生素治疗。(5) 纠正贫血和电解质紊乱: 输血、补充血制品 【人类医生评价】: 短时间失血过多大于全身血容量20%或者血红蛋白低于90可输血, 该患者不符合。 以及纠正低钙血症。
>  7. 治疗计划: (1) 急诊处理: 立即给予患者吸氧、心电监护, 建立静脉通路, 进行必要的输液、输血等支持治疗。(2) 完善腹部增强CT检查, 明确脾脏损伤程度及其他脏器损伤情况。(3) 根据CT结果, 如证实脾破裂严重, 立即安排急诊剖腹探查术, 术中根据脾脏损伤情况决定是否行脾切除或脾修补术, 并评估处理其他脏器损伤。(4) 在术前、术后给予广谱抗生素预防感染。(5) 根据血红蛋白和血细胞比容的变化, 适时输注红细胞悬液, 补充血容量。(6) 给予补钙治疗, 纠正低钙血症。(7) 术后严密监测患者生命体征、腹部体征及实验室指标变化, 调整治疗方案。
>  8. 影像诊断: (1) 肝脏: 形态大小正常, 被膜光滑连续, 实质回声细密、增强, 肝内血管结构清晰, 门静脉主干内径正常, 无异常发现。(2) 胆囊: 形态大小正常, 壁光滑, 囊腔内无异常回声, 肝内外胆管无扩张。(3) 脾脏: 形态大小正常, 实质回声不均匀, 脾脏与左肾间存在一片状不规则低回声区域, 边界欠清晰, 内部回声不均, 大小约为7.9×4.8cm, 提示可能存在脾脏局部病变, 如结节、肿瘤或其他占位性病变。(4) 胰腺: 形态大小正常, 实质回声均匀, 主胰管无扩张, 无异常发现。(5) 肾脏: 双肾位置、形态、大小正常, 轮廓光滑规整, 皮髓质界限清晰, 实质回声分布均匀, 集合系统无分离, 无积水现象。(6) CDFI: 血流信号未见明显异常。(7) 腹腔探查: 平卧位未见明显游离液性暗区, 提示无明显腹水。综上所述, 影像学诊断为: a. 脾脏实质回声不均匀, 脾脏与左肾间存在一个不规则低回声区, 建议进一步检查以明确该病灶的性质, 如CT、MRI或活组织病理检查等。b. 其他器官（肝脏、胆囊、胰腺、双肾及腹腔）未见明显异常。

Table 12: A detailed case study on ClinicalBench, diagnosed by Agent#3@1.

> **Case Study (Chinese)**
>
> 【Agent#1@1提供的科室导诊、临床诊断和影像诊断】：
>
> 1. 科室导诊: 肝胆胰外科。
>
> 2. 初步诊断: (1) 外伤性脾破裂伴血肿形成; (2) 脂肪肝; (3) 双侧肋骨骨折; (4) 轻度贫血; (5) 轻度低钙血症。
>
> 3. 诊断依据: (1) 患者有明确的车祸外伤史，尤其是撞击导致的 方向盘失控 【人类医生评价】：语序错误。 和双侧肋骨骨折。(2) 腹痛剧烈且持续不缓解，伴有腹肌紧张、压痛及反跳痛，提示可能存在腹内脏器损伤。(3) 彩色多普勒超声提示脾脏回声欠均匀，考虑挫裂伤可能，以及脾脏、左肾间的异常回声，局部包裹，考虑为血肿。(4) 实验室检查显示白细胞总数升高，中性粒细胞比例增加，C反应蛋白显著升高，提示存在急性炎症反应，同时血红蛋白和红细胞计数降低，提示轻度贫血。(5) 血生化结果显示总蛋白和白蛋白水平下降，钙离子浓度降低。
>
> 4. 鉴别诊断:
>
> 4.1. 胰腺损伤: 虽然超声未见明显腹腔积液，但胰腺损伤需进一步排查，因其症状与脾破裂相似，可通过CT等影像学检查予以排除。
>
> 4.2. 肝脏损伤: 超声提示脂肪肝，但未明确提及肝脏挫裂伤，需结合临床表现和进一步影像学检查以排除。
>
> 4.3. 小肠或结肠穿孔: 腹膜炎体征需与小肠或结肠穿孔相鉴别，通过CT或立位腹平片检查可帮助诊断。
>
> 5. 最终诊断: 外伤性脾破裂伴血肿形成。
>
> 6. 治疗原则: (1) 急诊手术治疗：对于脾破裂伴血肿形成的患者，若病情危重，应尽快行剖腹探查术，评估脾脏损伤情况，决定是否行脾切除或脾修补术。(2) 抗感染治疗：预防和控制感染，给予广谱抗生素治疗。(3) 纠正贫血和电解质紊乱：输血、补充血制品 【人类医生评价】：短时间失血过多大于全身血容量20%或者血红蛋白低于90可输血，该患者不符合。 以及纠正低钙血症。(4) 监测生命体征和器官功能：密切观察患者的生命体征、尿量、心电图、血气分析等，及时发现并处理可能出现的并发症。
>
> 7. 治疗计划: (1) 进一步完善腹部增强CT检查，明确脾脏损伤程度及是否存在其他脏器损伤。(2) 根据CT结果，如证实脾破裂严重，立即安排急诊剖腹探查术，术中根据脾脏损伤情况决定是否行脾切除或脾修补术。(3) 在术前、术后给予广谱抗生素预防感染。(4) 根据血红蛋白和血细胞比容的变化，适时输注红细胞悬液，补充血容量。(5) 给予补钙治疗，纠正低钙血症。(6) 术后严密监测患者生命体征、腹部体征及实验室指标变化，调整治疗方案。
>
> 8. 影像诊断: (1) 肝脏：正常形态和大小，被膜完整，实质回声细密增强，肝内血管结构清晰，门静脉主干内径在正常范围内。(2) 胆囊：形态和大小正常，壁光滑，囊腔内无异常回声，肝内外胆管未见扩张。(3) 脾脏：形态和大小正常，但实质回声不均匀，脾脏与左肾之间存在一不规则低回声区，边界不清，内部回声不均，大小约为7.9×4.8cm。(4) 胰腺：形态和大小正常，实质回声均匀，主胰管无扩张。(5) 双肾：位置、形态、大小正常，轮廓光滑规整，皮髓质界限清晰，实质回声分布均匀，集合系统无分离，无积水现象。(6) CDFI 血流信号检查未发现明显异常。(7) 平卧位探查腹腔，未见明显游离液性暗区，提示无明显腹水。综合以上影像表现，初步印象可能为脾脏局部病变（如结节、肿瘤等）以及脾脏与左肾间的不规则低回声区需要进一步评估。 【人类医生评价】：外伤后出血。 其他器官未见明显异常。

Table 13: A detailed case study on ClinicalBench, diagnosed by Agent#1@1.

> **Case Study (Chinese)**
>
> 【InternLM2提供的科室导诊、临床诊断和影像诊断】：
>  1. 科室导诊: 肝胆胰外科。
>  2. 初步诊断: (1) 腹部闭合性损伤：脾破裂、肝挫伤、肾挫伤；【人类医生评价】：病例中未提及。 (2) 腹腔内出血;(3) 脂肪肝;(4) 血小板减少症；【人类医生评价】：患者血小板减少是贫血所致，无需此诊断。 (5) 感染性休克。【人类医生评价】：患者无休克表现，休克有意识障碍，血压低于90/60mmHg。
>  3. 诊断依据: (1) 患者有明确的腹部外伤史，出现剧烈腹痛，腹肌紧张，有压痛及反跳痛，提示腹部闭合性损伤的可能。(2) 超声检查显示脾脏、左肾间异常回声，局部包裹，考虑血肿，提示脾破裂或肾挫伤的可能。(3) 实验室检查显示白细胞计数升高，中性粒细胞百分比和绝对值升高，提示感染的可能。(4) 血红蛋白和红细胞计数降低，提示存在失血性贫血。(5) 血小板计数降低，提示血小板减少症。【人类医生评价】：患者血小板减少是贫血所致，无需此诊断。 (6) 总蛋白、白蛋白和钙离子水平降低，提示存在低蛋白血症和电解质紊乱。
>  4. 鉴别诊断:
>  4.1. 肠梗阻：患者无恶心、呕吐等症状，腹部X线检查未见明显气液平，可排除肠梗阻。
>  4.2. 急性胰腺炎：患者无上腹部疼痛、恶心、呕吐等症状，血淀粉酶和脂肪酶水平正常，可排除急性胰腺炎。
>  4.3. 消化道穿孔：患者无气腹征象，腹部X线检查未见膈下游离气体，可排除消化道穿孔。
>  4.4. 腹膜炎：患者无发热、寒战等感染症状，腹部体征不典型，可排除腹膜炎。
>  5. 最终诊断: 腹部闭合性损伤（脾破裂、肝挫伤、肾挫伤【人类医生评价】：病例中无法证明有上述疾病。）。
>  6. 治疗原则: (1) 紧急处理：迅速评估患者生命体征，维持气道通畅，纠正休克，控制出血。(2) 手术治疗：对于脾破裂、肝挫伤、肾挫伤【人类医生评价】：病例中无法证明有上述疾病。等腹部闭合性损伤，应尽快进行手术治疗，清除坏死组织，止血，修复损伤器官。(3) 抗感染治疗：根据病原菌培养结果，选择合适的抗生素进行抗感染治疗。(4) 营养支持：给予患者足够的营养支持，纠正低蛋白血症和电解质紊乱。(5) 监测病情：密切监测患者生命体征、血常规、血生化等指标，及时调整治疗方案。
>  7. 治疗计划: (1) 立即进行腹部CT检查，明确脾破裂、肝挫伤、肾挫伤【人类医生评价】：病例中无法证明有上述疾病。的范围和程度。(2) 紧急输血、输液，纠正休克和失血性贫血。【人类医生评价】：未达到输血标准，患者无休克表现。 (3) 进行急诊手术，清除坏死组织，止血，修复损伤器官。(4) 术后给予广谱抗生素预防感染。(5) 给予患者肠内营养支持，纠正低蛋白血症和电解质紊乱。(6) 术后密切监测患者生命体征、血常规、血生化等指标，及时调整治疗方案。
>  8. 影像诊断: (1) 肝脏：形态、大小、被膜、实质回声、肝内管状结构均正常，门静脉主干内径正常。(2) 胆囊：形态、大小、壁光滑，囊腔内无异常回声，肝内外胆管无扩张。(3) 脾脏：形态、大小正常，实质回声不均匀，脾脏与左肾间可见不规则低回声区，边界欠清，内部回声不均，范围约7.9×4.8cm。(4) 胰腺：形态、大小、实质回声均匀，主胰管无扩张。(5) 肾脏：双肾位置、形态、大小、轮廓、皮髓质界限、实质回声分布均正常，集合系统无分离。(6) CDFI：血流信号未见明显异常。(7) 腹腔：平卧位探查，未见明显游离液性暗区。

Table 14: A detailed case study on ClinicalBench, diagnosed by InternLM2.

---

**Case Study (Chinese)**

【GPT-4提供的科室导诊、临床诊断和影像诊断】:

1. 科室导诊: 肝胆胰外科。

2. 初步诊断: (1) 腹部挫裂伤; 【人类医生评价】: 病例中未提及。 (2) 脾脏挫裂伤; (3) 脾脏血肿; 【人类医生评价】: 脾脏左肾间血肿。 (4) 肝脏挫伤; 【人类医生评价】: 病例中未提及。 (5) 脂肪肝; (6) 双侧肋骨骨折; (7) 腹膜炎。

3. 诊断依据: (1) 患者有车祸外伤史,腹部受到撞击。(2) 彩色多普勒超声显示脾脏、左肾间异常回声,局部包裹,考虑血肿,脾脏回声欠均匀,提示挫裂伤可能。(3) 彩色多普勒超声显示脂肪肝。(4) 患者有双侧肋骨骨折的明确诊断。(5) 腹肌紧张,有压痛及反跳痛,提示腹膜刺激征象。(6) 血常规显示白细胞升高,C反应蛋白升高,提示有炎症反应。(7) 血生化显示总蛋白、白蛋白降低,可能与腹部损伤导致的蛋白质丢失或稀释有关。

4. 鉴别诊断:

4.1. 胰腺挫裂伤: 需结合临床和影像学检查进一步排除。

4.2. 腹部器官 【人类医生评价】: 消化道。 穿孔: 患者无明显的胃肠道症状,如恶心、呕吐,且超声未见明显积液,但需警惕。

4.3. 腹部其他 实体 【人类医生评价】: 实质。 器官损伤: 需结合临床和影像学检查进一步排除。

5. 最终诊断: 脾脏挫裂伤。

6. 治疗原则: (1) 稳定患者生命体征。(2) 控制内出血,防止休克。(3) 预防和治疗感染。 【人类医生评价】: 保守治疗及手术后需抗生素治疗。 (4) 保护和恢复脾脏功能,避免不必要的脾切除。(5) 监测和处理可能的并发症。

7. 治疗计划: (1) 严密监测生命体征,维持血流动力学稳定。(2) 保守治疗: 卧床休息,禁食,静脉补液,输血治疗(如有必要)。(3) 抗生素治疗: 预防或治疗感染。(4) 疼痛控制: 适量使用镇痛药物。(5) 定期复查超声或CT,监测脾脏血肿和挫裂伤的变化。(6) 如出现脾脏破裂或持续出血,考虑手术治疗。(7) 针对脂肪肝,建议患者改善生活方式,调整饮食。(8) 对于肋骨骨折,给予适当的疼痛控制和支持性治疗。

8. 影像诊断: (1) 肝脏: 未见明显异常。 【人类医生评价】: 脂肪肝。 (2) 胆囊: 未见明显异常。(3) 脾脏: 实质回声不均匀,脾脏与左肾间存在一片状不规则低回声区,大小约7.9×4.8cm,需结合临床进一步评估(可能为脾周围病变或其他)。(4) 胰腺: 未见明显异常。(5) 肾脏: 未见明显异常。(6) 血流检查(CDFI): 未见明显异常。(7) 腹腔: 未见明显游离液体。

Table 15: A detailed case study on ClinicalBench, diagnosed by GPT-4.

Case Study (Chinese)

【Gemini-Pro提供的科室导诊、临床诊断和影像诊断】:

1. 科室导诊: 肝胆胰外科。

2. 初步诊断: (1) 脾脏挫裂伤; (2) 肝挫伤; 【人类医生评价】: 病例中未提及。 (3) 肋骨骨折; (4) 脂肪肝。

3. 诊断依据: (1) 患者有明确的车祸外伤史，且出现全腹痛，程度剧烈难以忍受，伴头痛不适，双侧肋骨骨折。(2) 影像学检查显示脾脏、左肾间异常回声，局部包裹，考虑血肿，较前范围变化不大；脾脏回声欠均匀，提示挫裂伤可能。(3) 实验室检查显示白细胞（WBC）升高，淋巴细胞百分数（LYMPH%）降低，中性粒细胞百分数（NEUT%）升高，单核细胞绝对值（MONO#）升高，中性粒细胞绝对值（NEUT#）升高，红细胞（RBC）降低，血红蛋白（HGB）降低，红细胞比容（HCT）降低，平均血小板容积（MPV）升高，嗜碱细胞绝对值（BASO#）升高，C反应蛋白（CRP）升高。(4) 血生化检查显示总蛋白（TP）降低，白蛋白（ALB）降低，钙（Ca）降低。

4. 鉴别诊断:

4.1. 胰腺挫伤：患者无恶心、呕吐，无意识不清，腹部平坦，无胃肠蠕动波，腹式呼吸存在，未见腹壁静脉曲张，肠鸣音减弱，余腹部查体因疼痛拒查。

4.2. 胆囊挫伤：患者无恶心、呕吐，无意识不清，腹部平坦，无胃肠蠕动波，腹式呼吸存在，未见腹壁静脉曲张，肠鸣音减弱，余腹部查体因疼痛拒查。【人类医生评价】：无需鉴别。

4.3. 肾挫伤：患者无血尿，无腰痛，无尿频、尿急、尿痛等症状。

5. 最终诊断: 脾脏挫裂伤。

6. 治疗原则: (1) 积极止血，纠正休克。【人类医生评价】：患者无休克表现。(2) 抗感染治疗。(3) 营养支持治疗。(4) 手术治疗。

7. 治疗计划: (1) 输血、输液，纠正休克。【人类医生评价】：未达到输血标准，患者无休克表现。(2) 抗生素治疗，预防感染。(3) 营养支持治疗，补充能量和蛋白质。(4) 根据患者病情，决定是否需要手术治疗。【人类医生补充】：(5) 需检测生命体征，建立静脉通路，及时补液治疗。

8. 影像诊断: 脾脏、左肾间可见一片状不规则低回声区，边界欠清，内部回声不均，范围约7.9×4.8cm。

Table 16: A detailed case study on ClinicalBench, diagnosed by Gemini-Pro.

---

**Case Study (Chinese)**

【Yi-34B-Chat提供的科室导诊、临床诊断和影像诊断】：
    1. 科室导诊: 肝胆胰外科。
    2. 初步诊断: (1) 腹部闭合性损伤; (2) 脾脏损伤; (3) 左肾损伤; 【人类医生评价】：病例中未提及。 (4) 脂肪肝; (5) 肋骨骨折; (6) 创伤性休克？（根据症状和血常规中的白细胞升高，不排除有休克的可能，但需要进一步检查确认） 【人类医生评价】：患者无休克表现。
    3. 诊断依据: (1) 腹部闭合性损伤：根据患者车祸外伤史和腹部疼痛、压痛及反跳痛的体征。(2) 脾脏损伤：彩色多普勒超声提示脾脏异常回声，考虑血肿形成，且脾脏回声欠均匀，提示挫裂伤可能。(3) 左肾损伤：彩色多普勒超声提示左肾间异常回声，考虑血肿形成。 【人类医生评价】：病例中未提及。 (4) 脂肪肝：彩色多普勒超声提示脂肪肝。(5) 肋骨骨折：根据病史中的方向盘失控和双侧肋骨骨折的描述。(6) 创伤性休克：根据症状和血常规中的白细胞升高，不排除有休克的可能，但需要进一步检查确认。 【人类医生评价】：患者无休克表现。
    4. 鉴别诊断:
    4.1. 肝破裂：虽然患者有腹部外伤和腹膜炎体征，但彩色多普勒超声未提示肝破裂，且肝浊音界存在，故暂不考虑肝破裂。
    4.2. 肠破裂：患者无明显腹膜炎症状，且彩色多普勒超声未提示肠管异常，故暂不考虑肠破裂。
    4.3. 胰腺损伤：患者无明显上腹痛、腰背痛及血淀粉酶升高等症状，彩色多普勒超声未提示胰腺异常，故暂不考虑胰腺损伤。
    5. 最终诊断: 腹部闭合性损伤（包括脾脏损伤、左肾损伤、 【人类医生评价】：病例中未提及。 肋骨骨折）。
    6. 治疗原则: (1) 紧急处理：包括保持呼吸道通畅、稳定生命体征、输液、输血等支持治疗。(2) 手术治疗：根据病情发展，可能需要进行剖腹探查术以明确损伤情况，并进行相应的修补或切除手术。(3) 非手术治疗：对于症状较轻、生命体征稳定的患者，可以先采取保守治疗，包括观察、输液、使用抗生素等。
    7. 治疗计划: (1) 急诊处理：立即给予患者吸氧、心电监护，建立静脉通路，进行必要的输液、输血等支持治疗。(2) 实验室检查：复查血常规、血生化、凝血功能等，监测血红蛋白、血小板、凝血酶原时间等指标的变化。(3) 影像学检查： 必要时进行腹部CT扫描， 【人类医生评价】：必须进行腹部CT扫描。 以进一步明确损伤部位和程度。(4) 手术准备：如果病情需要，应立即准备剖腹探查手术，包括准备手术室、麻醉、手术团队等。(5) 非手术治疗：对于生命体征稳定、症状较轻的患者，可以先采取保守治疗，包括卧床休息、观察生命体征、使用抗生素预防感染等。(6) 重症监护：对于有休克或其他严重并发症的患者，应转入重症监护室进行密切监测和治疗。请注意，以上诊断和治疗计划是基于提供的病例摘要和检查结果，实际情况可能需要根据患者的具体情况进行调整。由于缺乏病理学检查和一些关键的实验室检查结果，如凝血功能，上述诊断和治疗计划可能需要根据后续检查结果进行修正。此外，由于患者可能存在多器官损伤，治疗计划应根据损伤的严重程度和患者的整体状况进行综合考虑。
    8. 影像诊断: (1) 肝脏：形态大小正常，实质回声增强，门静脉主干内径正常，未见异常。(2) 胆囊：形态大小正常，壁光滑，囊腔内未见异常回声，肝内外胆管无扩张。(3) 脾脏：形态大小正常，实质回声不均匀，可见一片状低回声区，边界欠清，内部回声不均。(4) 胰腺：形态大小正常，实质回声均匀，主胰管无扩张。(5) 肾脏：双肾位置、形态、大小正常，实质回声分布均匀，集合系统无分离，未见异常。(6) CDFI：血流信号未见明显异常。(7) 腹腔探查：平卧位未见明显游离液性暗区。

Table 17: A detailed case study on ClinicalBench, diagnosed by Yi-34B-Chat.



**Case Study (English)**

[Case ID]: 94d8abab8a4643cc91c2443e96f00027

[Department]: Hepatobiliary & Pancreatic Surgery

[Case Summary]:
Patient Basic Information: Middle-aged male, XX years old. (We anonymize the age information in the sample data presented.)
Chief Complaint: Upper left abdominal pain for 2 hours.
Medical History: The patient developed severe and persistent upper left abdominal pain 2 hours ago after being struck by the steering wheel while driving. There are no factors that aggravate or relieve the pain. The patient does not exhibit headache, confusion, nausea, vomiting, chest tightness, shortness of breath, fever, fatigue, frequent urination, or urgency. There is evidence of bilateral rib fractures. The patient was brought to our emergency department by EMS. An abdominal ultrasound indicated "abnormal splenic echo suggestive of hematoma." The patient was admitted under the diagnosis of "traumatic splenic rupture" for further treatment. The patient has a clear history of trauma from a car accident. Previously healthy.
Physical Examination: The patient is alert but in distress and uncooperative during the examination. Coarse breath sounds and wet rales are audible in both lungs. Heart rhythm is regular without pathological murmurs. The abdomen is flat, without visible peristaltic waves, and abdominal breathing is present with no abdominal wall varices observed. The abdominal muscles are tense with significant tenderness and rebound tenderness in the upper left quadrant, along with muscle guarding. The liver dullness is intact, with the upper border of the liver located at the right midclavicular line in the fifth intercostal space. Bowel sounds are diminished. Further abdominal examination is refused due to pain. Muscle strength in the limbs is grade V, with normal muscle tone and no edema in the lower extremities.
Auxiliary Examination:
    1. Imaging Examination:
    1.1. Color Doppler Ultrasound: (1) Abnormal echogenicity between the spleen and left kidney, locally encapsulated, suggestive of hematoma. (2) Fatty liver. (3) Uneven echogenicity of the spleen, indicating a possible contusion. (4) No obvious effusion is seen in the visualized parts of the abdominal cavity; please correlate with clinical findings and consider routine ultrasound follow-up if necessary.
    2. Laboratory Examination:
    2.1. Routine Blood Test: (1) White Blood Cells (WBC) $10.8 \times 10^9/L$ ↑; (2) Lymphocyte Percentage (LYMPH%) 11.9% ↓; (3) Neutrophil Percentage (NEUT%) 78.7% ↑; (4) Monocyte Count (MONO#) $0.64 \times 10^9/L$ ↑; (5) Neutrophil Count (NEUT#) $8.5 \times 10^9/L$ ↑; (6) Red Blood Cells (RBC) $3.4 \times 10^{12}/L$ ↓; (7) Haemoglobin (HGB) $104g/L$ ↓; (8) Hematocrit (HCT) 31.0% ↓; (9) Mean Platelet Volume (MPV) $10.1fL$ ↑; (10) Basophils Count (BASO#) $0.07 \times 10^9/L$ ↑; (11) C-reactive Protein (CRP) $45.77mg/L$ ↑.
    2.2. Blood Biochemistry Test: (1) Total Protein (TP) $55.4g/L$ ↓; (2) Albumin (ALB) $35.1g/L$ ↓; (3) Calcium (Ca) $2.06mmol/L$ ↓.

[Color Doppler Ultrasound]:
    Findings: (1) Liver: Normal in shape and size, with a smooth and continuous capsule. The parenchymal echoes are fine and enhanced. The intrahepatic tubular structures are clearly and naturally aligned, and the main lumen of the portal vein is of normal diameter. (2) Gallbladder: Normal in shape and size, with a still smooth wall. No abnormal echoes are seen in the lumen, and there is no dilatation of the intrahepatic and extrahepatic bile ducts. (3) Spleen: Normal in shape and size, with uneven parenchymal echoes. Between the spleen and left kidney, there is an irregularly shaped hypoechoic area with unclear boundaries and uneven internal echoes, measuring approximately 7.9×4.8 cm. (4) Pancreas: Normal in shape and size, with uniform parenchymal echoes, and no dilatation of the main pancreatic duct. (5) Kidneys: Both kidneys are normal in position, shape, and size, with smooth and regular contours, clear corticomedullary differentiation, and uniform distribution of parenchymal echoes. The collecting system shows no separation. (6) CDFI: No significant abnormalities in blood flow signals. Supine examination of the abdominal cavity revealed no significant free fluid dark areas in the visible parts.



Table 18: A detailed case study on ClinicalBench.



**Case Study (English)**

[Clinical Diagnosis & Imaging Diagnosis Provided by Doctor]:

1. Preliminary Diagnosis: (1) Traumatic splenic rupture; (2) Rib fractures; (3) Traumatic pneumonia.

2. Diagnostic Basis:

2.1. The patient experienced upper left abdominal pain 2 hours after a trauma.

2.2. Physical examination supports the diagnosis: The patient is alert with a painful expression and shows poor cooperation during the examination. Coarse respiratory sounds in both lungs with audible moist sounds, regular heart rhythm with no pathological murmurs heard, flat abdomen with no gastrointestinal peristalsis waves, abdominal respiration present, no visible abdominal wall varicose veins. Tense abdominal muscles with significant tenderness in the upper left abdomen, no rebound tenderness or muscle rigidity noted, liver dullness present with the upper liver border at the right midclavicular line at the level of the fifth rib, reduced bowel sounds, remainder of the abdominal examination not conducted due to pain. Limb strength grade V, normal muscle tone, and no swelling in the lower extremities.

2.3. Imaging examinations support the diagnosis: (1) Color Doppler Ultrasound shows abnormal echoes between the spleen and left kidney, suggesting a localized encapsulated hematoma, with little change from previous; fatty liver; uneven spleen echoes suggest a contusion injury; part of the abdominal cavity shows no significant fluid accumulation. Please correlate with clinical findings and consider routine ultrasound follow-up if necessary.

2.4. Laboratory examinations support the diagnosis: (1) Routine Blood Test shows elevated white blood cells (WBC), decreased lymphocyte percentage (LYMPH%), increased neutrophil percentage (NEUT%), elevated monocyte count (MONO#), increased neutrophil count (NEUT#), decreased red blood cells (RBC), decreased hemoglobin (HGB), decreased hematocrit (HCT), elevated mean platelet volume (MPV), elevated basophil count (BASO#), and increased C-reactive protein (CRP). (2) Blood biochemistry Test shows decreased total protein (TP), decreased albumin (ALB), and decreased calcium (Ca).

3. Differential Diagnosis:

3.1. Acute perforation of peptic ulcer: Typical history of ulcer disease, sudden worsening of abdominal pain, tense abdominal muscles, disappearance of liver dullness, and the presence of free gas under the diaphragm seen on X-ray.

3.2. Cholelithiasis and acute cholecystitis: Typically presents with biliary colic, pain located in the upper right abdomen radiating to the right shoulder, positive Murphy's sign, slight elevation of blood and urine amylase, with ultrasound and X-ray cholangiography confirming the diagnosis.

3.3. Acute necrotizing enteritis: Presents with abdominal pain, distension, bloody stools, high fever, and vomiting, with severe abdominal distension and significant systemic toxic symptoms.

4. Final Diagnosis: Splenic rupture.

5. Principle of Treatment: (1) Prioritize life over spleen preservation; (2) Younger age favors spleen preservation; (3) Choose one or several spleen-preserving methods based on the extent of splenic damage; (4) Close monitoring is necessary after spleen-preserving surgery to prevent delayed splenic rupture; (5) For elderly patients, those in poor general condition, severe multiple injuries, or significant prolongation of prothrombin time, splenectomy is recommended.

6. Treatment Plan: (1) Provide critical care, cardiac monitoring, establish venous access, declare medical emergency, and closely monitor vital signs. (2) Administer fluids to restore blood volume. (3) Determine surgical indications, exclude contraindications, and proceed with emergency surgery.

7. Impression: (1) Abnormal echoes between the spleen and left kidney, suggesting a localized encapsulated hematoma, with little change from previous. (2) Fatty liver. (3) Uneven spleen echoes suggest a contusion injury. (4) Part of the abdominal cavity shows no significant fluid accumulation; please correlate with clinical findings and consider routine ultrasound follow-up if necessary.



Table 19: A detailed case study on ClinicalBench, diagnosed by human doctor.

**Case Study (English)**

[Clinical Diagnosis & Imaging Diagnosis Provided by Agent#1@3]:

1. Department Guide: Hepatobiliary & Pancreatic Surgery.

2. Preliminary Diagnosis: (1) Traumatic splenic rupture with hematoma formation; (2) Closed abdominal trauma; (3) Bilateral rib fractures; (4) Mild anemia; (5) Mild hypocalcemia; (6) Fatty liver.

3. Diagnostic Basis: (1) The patient has a clear history of trauma from a car accident, particularly severe upper left abdominal pain and left rib fractures resulting from impact with the steering wheel. (2) Severe abdominal pain, muscular tension, tenderness, and rebound tenderness suggest possible internal organ injury. (3) Color Doppler Ultrasound shows uneven echogenicity of the spleen, suggesting a possible contusion, and abnormal echos between the spleen and left kidney, enclosed, suggesting a hematoma. (4) Laboratory examinations show elevated white blood cells and neutrophil percentage, significantly increased C-reactive protein indicating an acute inflammatory response; decreased hemoglobin and red blood cell suggest mild anemia; reduced calcium levels indicate mild hypocalcemia. (5) Ultrasound did not reveal significant abdominal effusion but does not entirely rule out liver contusion and other visceral injuries.

4. Differential Diagnosis:

4.1. Pancreatic injury: Although no significant abdominal effusion was seen on ultrasound, pancreatic injury needs further exclusion due to symptoms similar to splenic rupture, which can be excluded via CT or other imaging tests.

4.2. Intestinal rupture or perforation: Signs of peritonitis need to be differentiated from small or large intestine perforation, which can be diagnosed with CT or upright abdominal plain films.

4.3. Kidney injury: Ultrasound suggests abnormal echos between the spleen and left kidney, further investigation is needed to ascertain kidney damage.

4.4. Thrombocytopenia: Although platelet count is within normal range, increased MPV suggests possible platelet dysfunction rather than a decrease in quantity. [Human Doctor Comment]: Intra--abdominal bleeding has led to anemia, which does not require differentiation from hematologic diseases.

5. Final Diagnosis: Traumatic splenic rupture with hematoma formation.

6. Principle of Treatment: (1) Emergency management: Assess vital signs, maintain airway patency, correct shock, control bleeding. (2) Surgical treatment: For splenic rupture with hematoma formation, surgical intervention should be undertaken promptly to assess the spleen's damage and decide whether splenectomy or splenic repair is necessary. [Human Doctor Comment]: If the spleen rupture is not severe, it can be managed with strict bed rest and observation. (3) Anti-infection treatment: Prevent and control infection, administer broad-spectrum antibiotics. (4) Correct anemia and electrolyte disorder: Transfusion, blood products supplementation, and correction of hypocalcemia. (5) Monitor vital signs and organ functions: Close observation of vital signs, urine output, ECG, blood gas analysis, etc., to promptly detect and manage potential complications.

7. Treatment Plan: (1) Complete abdominal CT scan to clarify the extent of splenic damage, the presence of other organ injuries (like liver contusion), and kidney condition. (2) If splenic rupture is confirmed as severe, schedule an emergency exploratory laparotomy, decide during surgery whether to perform splenectomy or splenic repair, and address other potential organ injuries. (3) Administer broad-spectrum antibiotics pre- and post-surgery to prevent infection. (4) Based on changes in hemoglobin and hematocrit, timely transfuse red blood cell suspension to supplement blood volume. (5) Administer calcium treatment to correct hypocalcemia. (6) Post-surgery, closely monitor patient's vital signs, abdominal signs, and laboratory index changes, adjusting the treatment plan accordingly. If liver contusion is severe, corresponding measures need to be taken. [Human Doctor Comment]: The diagnosis does not indicate liver contusion.

Table 20: A detailed case study on ClinicalBench, diagnosed by Agent#1@3.

> ### Case Study (English)
>
> 8. Impression: (1) Normal morphology, size, structure, blood flow signals of the liver, gallbladder, pancreas, and both kidneys; intrahepatic and extrahepatic bile ducts, the main trunk of the portal vein, and collecting system appear normal. (2) However, the parenchyma of the spleen shows heterogeneous echogenicity changes, and a patchy, irregular hypoechoic area between the spleen and left kidney, with unclear boundaries and uneven internal echoes, measuring about 7.9×4.8cm. This finding suggests the possibility of a local space-occupying lesion in the spleen, such as nodules, tumors, or other inflammatory lesions. Further investigation with CT, MRI, or tissue biopsy is recommended. (3) No significant free fluid shadows observed within the abdominal cavity, ruling out the possibility of large volume ascites. In summary, imaging shows local abnormality in the spleen and irregular hypoechoic area between the spleen and kidney, necessitating further investigation of the space-occupying lesion. No significant abnormalities observed in other related organs.

Table 21: A detailed case study on ClinicalBench, diagnosed by Agent#1@3. (Cont. Table20)

## Case Study (English)

[Clinical Diagnosis & Imaging Diagnosis Provided by Agent#3@1]:

1. Department Guide: Hepatobiliary & Pancreatic Surgery.

2. Preliminary Diagnosis: (1) Traumatic splenic rupture with hematoma formation; (2) ==Left kidney contusion and laceration;== [Human Doctor Comment]: The Color Doppler Ultrasound did not indicate renal damage. (3) Fatty liver; (4) Bilateral rib fractures; (5) Mild anemia; (6) Mild hypocalcemia.

3. Diagnostic Basis: (1) The patient has a clear history of a traffic accident, specifically an impact that caused the steering wheel to lose control and bilateral rib fractures. (2) Severe abdominal pain, abdominal muscle tension, tenderness, and rebound tenderness suggest potential internal organ injury. (3) Color Doppler Ultrasound indicates uneven splenic echogenicity, suggesting possible contusion, and abnormal echoes between the spleen and left kidney with localized encapsulation, indicating hematoma formation. (4) Laboratory examinations show elevated white blood cells, increased neutrophil percentage, significantly elevated C-reactive protein indicating acute inflammatory response; reduced hemoglobin and red blood cells indicating mild anemia; decreased blood calcium levels. (5) Ultrasound shows no significant abdominal effusion, liver contusion not specifically mentioned, and the liver dullness border is present, suggesting the liver may not be severely damaged.

4. Differential Diagnosis:

4.1. Pancreatic injury: Although no significant abdominal effusion was seen on ultrasound, further investigation for pancreatic injury is needed, which can be ruled out by imaging studies such as CT.

4.2. Liver contusion: Ultrasound indicates fatty liver, but liver contusion is not specifically mentioned, requiring correlation with clinical presentation and further imaging studies for exclusion.

4.3. Intestinal rupture: The patient shows no obvious symptoms of peritonitis, and ultrasound does not suggest intestinal abnormalities, therefore, intestinal rupture is not considered.

4.4. Gallbladder injury: The patient lacks typical symptoms of gallbladder injury such as jaundice and upper right abdominal pain, therefore, not considered.

4.5. Kidney contusion and laceration: Ultrasound indicates ==abnormal echoes between the kidneys,== [Human Doctor Comment]: Abnormal echoes between the spleen and the kidney, rather than abnormal renal echoes. suggesting hematoma formation, thus included in the preliminary diagnosis.

5. Final Diagnosis: Traumatic splenic rupture with hematoma formation.

6. Principle of Treatment: (1) Emergency management: Maintain vital signs stability, including oxygen therapy, ECG monitoring, establishing intravenous access, fluid resuscitation, and blood transfusion. (2) Surgical treatment: Depending on the condition assessment, exploratory laparotomy may be needed to determine the extent of damage and perform splenectomy or splenic repair while assessing and managing other organ injuries. (3) Non-surgical treatment: For patients with mild symptoms and stable vital signs, conservative treatment may be attempted, including observation, fluid resuscitation, and antibiotic therapy. (4) Anti-infection treatment: Prevent and control infection, administer broad-spectrum antibiotics. (5) Correction of anemia and electrolyte disturbances: ==Transfusion of blood products== [Human Doctor Comment]: Transfusion is indicated for acute blood loss exceeding 20% of total blood volume or when hemoglobin levels fall below 90 g/L; this patient does not meet the criteria. and correction of hypocalcemia.

7. Treatment Plan: (1) Emergency management: Immediately provide the patient with oxygen therapy, ECG monitoring, establish intravenous access, and necessary support including fluid resuscitation and blood transfusion. (2) Complete abdominal enhanced CT scan to determine the extent of splenic injury and other organ injuries. (3) Based on CT results, if severe splenic rupture is confirmed, immediately arrange emergency exploratory laparotomy, during which the extent of splenic injury will dictate whether splenectomy or splenic repair is performed, and other organ injuries are assessed. (4) Administer broad-spectrum antibiotics preoperatively and postoperatively to prevent infection. (5) Based on changes in hemoglobin and hematocrit, timely transfusion of red cell suspensions to replenish blood volume. (6) Administer calcium supplementation to correct hypocalcemia. (7) Postoperatively, closely monitor the patient's vital signs, abdominal signs, and laboratory indicators to adjust the treatment plan.

Table 22: A detailed case study on ClinicalBench, diagnosed by Agent#3@1.

2052
2053
2054
2055
2056
2057
2058
2059
2060
2061
2062
2063
2064
2065
2066
2067
2068
2069
2070
2071
2072
2073
2074
2075
2076
2077
2078
2079
2080
2081
2082
2083
2084
2085
2086
2087
2088
2089
2090
2091
2092
2093
2094
2095
2096
2097
2098
2099
2100
2101
2102
2103
2104
2105

---

**Case Study (English)**

8. Impression: (1) Liver: Normal shape and size, smooth continuous capsule, fine, enhanced parenchymal echogenicity, clear intrahepatic vascular structures, normal main portal vein diameter, no abnormalities. (2) Gallbladder: Normal shape and size, smooth wall, no abnormal echoes within the lumen, no dilation of intrahepatic and extrahepatic bile ducts. (3) Spleen: Normal shape and size, uneven parenchymal echoes, an irregular hypoechoic area between the spleen and left kidney, unclear boundaries, uneven internal echoes, measuring about 7.9×4.8 cm, suggesting possible localized lesion, such as nodules, tumors, or other space-occupying lesions. (4) Pancreas: Normal shape and size, uniform parenchymal echoes, no dilation of the main pancreatic duct, no abnormalities. (5) Kidneys: Normal position, shape, size, smooth contours, clear corticomedullary differentiation, uniform parenchymal echo distribution, no separation of the collecting system, no hydronephrosis. (6) CDFI: No significant abnormalities in blood flow signals. (7) Abdominal Exploration: In supine position, no significant free fluid dark areas seen, suggesting no significant ascites. Overall impression: a. Uneven spleen parenchymal echoes, irregular hypoechoic area between the spleen and left kidney, further examination recommended to clarify the nature of the lesion, such as CT, MRI, or live tissue pathology. b. No significant abnormalities in other organs (liver, gallbladder, pancreas, kidneys, and abdomen).

Table 23: A detailed case study on ClinicalBench, diagnosed by Agent#3@1. (Cont. Table22)

### Case Study (English)

[Clinical Diagnosis & Imaging Diagnosis Provided by Gemini-Pro]:

1. Department Guide: Hepatobiliary & Pancreatic Surgery.

2. Preliminary Diagnosis: (1) Splenic laceration; (2) Liver contusion; [Human Doctor Comment]: Not mentioned in the case. (3) Rib fractures; (4) Fatty liver.

3. Diagnostic Basis: (1) The patient has a clear history of trauma from a car accident, presenting with severe, unbearable abdominal pain accompanied by headaches and bilateral rib fractures. (2) Imaging examinations show abnormal echoes between the spleen and left kidney, locally encapsulated, suggesting hematoma with no significant change from previous observations; uneven echo from the spleen suggests possible laceration. (3) Laboratory examinations indicate elevated white blood cells (WBC), decreased lymphocyte percentage (LYMPH%), increased neutrophil percentage (NEUT%), elevated monocyte count (MONO#), increased neutrophil count (NEUT#), decreased red blood cells (RBC), decreased hemoglobin (HGB), decreased hematocrit (HCT), elevated mean platelet volume (MPV), elevated basophil count (BASO#), and increased C-reactive protein (CRP). (4) Blood Biochemistry Tests show decreased total protein (TP), decreased albumin (ALB), and decreased calcium (Ca).

4. Differential Diagnosis:

4.1. Pancreatic contusion: No nausea, vomiting, or loss of consciousness; the abdomen is flat without visible peristaltic waves, abdominal breathing is present, no abdominal wall varices observed, bowel sounds are diminished, and further abdominal examination was refused due to pain.

4.2. Gallbladder contusion: No nausea, vomiting, or loss of consciousness; the abdomen is flat without visible peristaltic waves, abdominal breathing is present, no abdominal wall varices observed, bowel sounds are diminished, and further abdominal examination was refused due to pain. [Human Doctor Comment]: No differentiation required.

4.3. Renal contusion: No hematuria, no back pain, and no symptoms of frequent urination, urgency, or pain during urination.

5. Final Diagnosis: Splenic laceration.

6. Principle of Treatment: (1) Aggressive hemostasis and shock correction. [Human Doctor Comment]: The patient shows no signs of shock. (2) Anti-infection treatment.(3) Nutritional support therapy. (4) Surgical treatment if necessary.

7. Treatment Plan: (1) Blood transfusion and fluid therapy to correct shock. [Human Doctor Comment]: The patient does not meet the criteria for transfusion and shows no signs of shock. (2) Antibiotic treatment to prevent infection. (3) Nutritional support therapy to provide energy and protein. (4) Surgery may be considered depending on the patient's condition. [Human Doctor Supplement]: (5) It is necessary to monitor vital signs, establish venous access, and provide timely fluid resuscitation.

8. Impression: A sheet-like irregular hypoechoic area is visible between the spleen and left kidney, with unclear boundaries and uneven internal echoes, measuring approximately 7.9×4.8 cm.

Table 24: A detailed case study on ClinicalBench, diagnosed by Gemini-Pro.

---

**Case Study (English)**

[Clinical Diagnosis & Imaging Diagnosis Provided by Agent#1@1]:

    1. Department Guide: Hepatobiliary & Pancreatic Surgery.

    2. Preliminary Diagnosis: (1) Traumatic splenic rupture with hematoma formation; (2) Fatty liver; (3) Bilateral rib fractures; (4) Mild anemia; (5) Mild hypocalcemia.

    3. Diagnostic Basis: (1) The patient has a clear history of trauma from a car accident, particularly the ==steering wheel== ==[Human Doctor Comment]: Incorrect word order.== impact and bilateral rib fractures. (2) Severe and persistent abdominal pain accompanied by abdominal muscle tension, tenderness, and rebound pain, suggesting possible visceral organ damage. (3) Color Doppler Ultrasound indicates uneven echogenicity of the spleen, suggestive of a contusion, and abnormal echogenicity between the spleen and left kidney, localized encapsulation, suggesting hematoma. (4) Laboratory examinations show increased total white blood cells, increased percentage of neutrophils, and significantly elevated C-reactive protein, indicating an acute inflammatory response; simultaneously, reduced hemoglobin and red blood cells suggest mild anemia. (5) Blood Biochemistry Test shows decreased total protein and albumin levels, and reduced calcium ion concentration.

    4. Differential Diagnosis:

    4.1. Pancreatic Injury: Although ultrasound shows no obvious abdominal effusion, further investigation is needed for pancreatic injury, as its symptoms are similar to splenic rupture, which can be excluded by imaging studies such as CT.

    4.2. Liver Injury: Ultrasound indicates fatty liver, but liver contusion is not specifically mentioned; further imaging studies are needed to exclude this, combined with clinical symptoms.

    4.3. Small intestine or colon perforation: Peritonitis signs need to be differentiated from small intestine or colon perforation, which can be diagnosed with the help of CT or upright abdominal plain films.

    5. Final Diagnosis: Traumatic splenic rupture with hematoma formation.

    6. Principle of Treatment: (1) Emergency surgery: For patients with splenic rupture and hematoma formation, if the condition is critical, exploratory laparotomy should be performed as soon as possible to assess the damage to the spleen and decide whether to perform a splenectomy or splenorrhaphy. (2) Anti-infection treatment: To prevent and control infection, broad-spectrum antibiotics are administered. (3) Correction of anemia and electrolyte disorders: ==Blood transfusions, blood product supplementation,== ==[Human Doctor Comment]: Transfusion is indicated for acute blood loss exceeding 20% of total blood volume or when hemoglobin levels fall below 90 g/L; this patient does not meet the criteria.== and correction of hypocalcemia. (4) Monitoring of vital signs and organ function: Close observation of the patient's vital signs, urine output, ECG, blood gas analysis, etc., to promptly detect and manage possible complications.

    7. Treatment Plan: (1) Further refine abdominal enhanced CT scans to clarify the extent of splenic injury and the presence of other organ damage. (2) Based on CT results, if splenic rupture is severe, arrange for emergency exploratory laparotomy, during which the extent of spleen damage will determine whether a splenectomy or splenorrhaphy is performed. (3) Administer broad-spectrum antibiotics before and after surgery to prevent infection. (4) Based on changes in hemoglobin and hematocrit, timely transfusion of red blood cells to replenish blood volume. (5) Calcium supplementation to correct hypocalcemia. (6) Post-surgery, closely monitor the patient's vital signs, abdominal signs, and laboratory indicator changes to adjust treatment plans.

Table 25: A detailed case study on ClinicalBench, diagnosed by Agent#1@1.

> **Case Study (English)**
>
> 8. Impression: (1) Liver: Normal shape and size, intact capsule, enhanced homogeneous echogenicity, clear intrahepatic vascular structures, normal main portal vein diameter. (2) Gallbladder: Normal shape and size, smooth wall, no abnormal echoes in the cavity, no dilation seen in intra- and extrahepatic bile ducts. (3) Spleen: Normal shape and size, but uneven echogenicity, an irregular hypoechoic area between the spleen and left kidney, unclear boundaries, inhomogeneous internal echoes, approximately 7.9×4.8 cm in size. (4) Pancreas: Normal shape and size, uniform echogenicity, no dilation of the main pancreatic duct. (5) Kidneys: Normal position, shape, size, smooth and regular contours, clear corticomedullary differentiation, uniform distribution of parenchymal echoes, no separation in the collecting system, no hydronephrosis. (6) CDFI blood flow signal examination found no significant abnormalities. (7) Supine position exploration of the abdominal cavity showed no obvious free liquid dark areas, suggesting no significant ascites. Based on the above imaging findings, the initial impression might be a local splenic lesion (such as nodules, tumors, etc.) and the irregular hypoechoic area between the spleen and left kidney needs further evaluation. [Human Doctor Comment]: Post-traumatic bleeding. No obvious abnormalities were seen in other organs.

Table 26: A detailed case study on ClinicalBench, diagnosed by Agent#1@1. (Cont. Table25)

> ### Case Study (English)
>
> [Clinical Diagnosis & Imaging Diagnosis Provided by GPT-4]:
>     1. Department Guide: Hepatobiliary & Pancreatic Surgery.
>     2. Preliminary Diagnosis: (1) Abdominal contusion; [Human Doctor Comment]: Not mentioned in the case. (2) Splenic contusion; (3) Splenic hematoma; [Human Doctor Comment]: Hematoma between the spleen and left kidney. (4) Hepatic contusion; [Human Doctor Comment]: Not mentioned in the case. (5) Fatty liver; (6) Bilateral rib fractures; (7) Peritonitis.
>     3. Diagnostic Basis: (1) The patient has a history of a traffic accident with abdominal impact. (2) Color Doppler Ultrasound shows abnormal echoes between the spleen and left kidney, local encapsulation suggesting hematoma, and uneven splenic echoes indicating possible contusion. (3) Color Doppler Ultrasound indicates fatty liver. (4) There is a confirmed diagnosis of bilateral rib fractures. (5) Abdominal muscle tension, with tenderness and rebound pain, indicating signs of peritoneal irritation. (6) Routine Blood Test shows increased white blood cells and C-reactive protein, suggesting an inflammatory response. (7) Blood Biochemistry Test shows decreased total protein and albumin, possibly related to protein loss or dilution due to abdominal injury.
>     4. Differential Diagnosis:
>     4.1. Pancreatic contusion: Requires further exclusion based on clinical and imaging examinations.
>     4.2. Abdominal organ [Human Doctor Comment]: Digestive tract. perforation: The patient shows no obvious gastrointestinal symptoms such as nausea or vomiting, and ultrasound does not reveal significant effusion, but caution is needed.
>     4.3. Damage to other solid abdominal organs: Further exclusion is required based on clinical and imaging examinations.
>     5. Final Diagnosis: Splenic contusion.
>     6. Principle of Treatment: (1) Stabilize the patient's vital signs. (2) Control internal bleeding to prevent shock. (3) Prevent and treat infections. [Human Doctor Comment]: Antibiotic treatment is required after conservative treatment and surgery. (4) Protect and restore splenic function, avoiding unnecessary splenectomy. (5) Monitor and manage potential complications.
>     7. Treatment Plan: (1) Closely monitor vital signs and maintain hemodynamic stability. (2) Conservative treatment: bed rest, fasting, intravenous fluid replenishment, and transfusion therapy (if necessary). (3) Antibiotic treatment: to prevent or treat infections. (4) Pain control: appropriate use of analgesics. (5) Regular re-evaluation by ultrasound or CT to monitor changes in splenic hematoma and contusion. (6) If splenic rupture or persistent bleeding occurs, consider surgical treatment. (7) For fatty liver, advise the patient to improve lifestyle and adjust diet. (8) For rib fractures, provide appropriate pain control and supportive treatment.
>     8. Impression: (1) Liver: No significant abnormalities. [Human Doctor Comment]: Fatty liver. (2) Gallbladder: No significant abnormalities. (3) Spleen: Heterogeneous parenchymal echoes with an irregular hypoechoic area between the spleen and left kidney, approximately 7.9×4.8cm, further clinical correlation needed (possible perisplenic lesion or other). (4) Pancreas: No significant abnormalities. (5) Kidneys: No significant abnormalities. (6) Blood flow examination (CDFI): No significant abnormalities. (7) Abdominal cavity: No significant free fluid.

Table 27: A detailed case study on ClinicalBench, diagnosed by GPT-4.

### Case Study (English)

[Clinical Diagnosis & Imaging Diagnosis Provided by InternLM2]:

1. Department Guide: Hepatobiliary & Pancreatic Surgery.

2. Preliminary Diagnosis: (1) Abdominal blunt trauma: splenic rupture, liver contusion, kidney contusion; [Human Doctor Comment]: Not mentioned in the case. (2) Intraperitoneal hemorrhage; (3) Fatty liver; (4) Thrombocytopenia; [Human Doctor Comment]: The patient's thrombocytopenia is due to anemia; no further diagnosis is needed. (5) Septic shock. [Human Doctor Comment]: The patient shows no signs of shock. Shock is characterized by disturbances in consciousness and blood pressure lower than 90/60 mmHg.

3. Diagnostic Basis: (1) The patient has a clear history of abdominal trauma, presenting with severe abdominal pain, abdominal muscle tension, tenderness, and rebound pain, suggesting the possibility of abdominal blunt trauma. (2) Ultrasonography shows abnormal echos between the spleen and left kidney, with localized encapsulation, considering hematoma, suggesting the possibility of splenic rupture or kidney contusion. (3) Laboratory examinations show increased white blood cells, increased percentage and count of neutrophils, suggesting the possibility of infection. (4) Decreased hemoglobin and red blood cells, indicating hemorrhagic anemia. (5) Decreased mean platelet volume, indicating thrombocytopenia. [Human Doctor Comment]: The patient's thrombocytopenia is due to anemia; no further diagnosis is needed. (6) Decreased total protein, albumin, and calcium levels, indicating hypoalbuminemia and electrolyte disturbances.

4. Differential Diagnosis:

4.1. Intestinal obstruction: The patient shows no symptoms of nausea or vomiting, and abdominal X-rays do not reveal any apparent air-fluid levels, ruling out intestinal obstruction.

4.2. Acute pancreatitis: The patient has no upper abdominal pain, nausea, or vomiting, and levels of serum amylase and lipase are normal, ruling out acute pancreatitis.

4.3. Gastrointestinal perforation: The patient shows no signs of pneumoperitoneum, and abdominal X-rays do not reveal free air under the diaphragm, ruling out gastrointestinal perforation.

4.4. Peritonitis: The patient shows no signs of fever or chills, and abdominal signs are not typical, ruling out peritonitis.

5. Final Diagnosis: Abdominal blunt trauma (splenic rupture, liver contusion, kidney contusion [Human Doctor Comment]: The case does not provide evidence of the aforementioned disease. ).

6. Principle of Treatment: (1) Emergency management: quickly assess patient vital signs, maintain airway patency, correct shock, and control bleeding. (2) Surgical treatment: For abdominal blunt trauma such as splenic rupture, liver contusion, and kidney contusion, [Human Doctor Comment]: The case does not provide evidence of the aforementioned disease. surgery should be performed as soona s possible to remove necrotic tissue, stop bleeding, and repair damaged organs. (3) Anti-infection treatment: Select appropriate antibiotics based on culture results to treat infections. (4) Nutritional support: Provide sufficient nutritional support to the patient to correct hypoalbuminemia and electrolyte disturbances. (5) Monitor the condition: Closely monitor patient vital signs, blood routine, and biochemistry indicators, and adjust the treatment plan timely.

7. Treatment Plan: (1) Immediate abdominal CT scan to clarify the extent and severity of splenic rupture, liver contusion, and kidney contusion. [Human Doctor Comment]: The case does not provide evidence of the aforementioned disease. (2) Emergency blood transfusion and fluid resuscitation to correct shock and hemorrhagic anemia. [Human Doctor Comment]: The patient does not meet the criteria for transfusion and shows no signs of shock. (3) Emergency surgery to remove necrotic tissue, stop bleeding, and repair damaged organs. (4) Postoperative administration of broad-spectrum antibiotics to prevent infection. (5) Provide enteral nutrition support to the patient to correct hypoalbuminemia and electrolyte disturbances. (6) Closely monitor postoperative patient vital signs, blood routine, and biochemistry indicators, and adjust the treatment plan timely.

Table 28: A detailed case study on ClinicalBench, diagnosed by InternLM2.

Case Study (English)

8. Impression: (1) Liver: Normal morphology, size, capsule, parenchymal echogenicity, and intrahepatic tubular structures; the main portal vein diameter is normal. (2) Gallbladder: Normal morphology, size, smooth walls, no abnormal echoes within the lumen, no dilation of intra- or extra-hepatic bile ducts. (3) Spleen: Normal morphology and size, uneven parenchymal echoes, irregular hypoechoic area between the spleen and left kidney, indistinct margins, inhomogeneous internal echoes, approximately $7.9 \times 4.8$ cm. (4) Pancreas: Normal morphology, size, and parenchymal echos, no dilation of the main pancreatic duct. (5) Kidneys: Normal position, morphology, size, outline, corticomedullary differentiation, and parenchymal echo distribution, no separation in the collecting system. (6) CDFI: No significant abnormalities in blood flow signals detected. (7) Abdominal cavity: Supine position exploration, no apparent free fluid dark areas observed.

Table 29: A detailed case study on ClinicalBench, diagnosed by InternLM2. (Cont. Table28)

> **Case Study (English)**
>
> [Clinical Diagnosis & Imaging Diagnosis Provided by Yi-34B-Chat]:
>
> 1. Department Guide: Hepatobiliary & Pancreatic Surgery.
>
> 2. Preliminary Diagnosis: (1) Closed abdominal trauma; (2) Splenic injury; (3) Left kidney injury; [Human Doctor Comment]: Not mentioned in the case. (4) Fatty liver; (5) Rib fracture; (6) Traumatic shock? (Based on symptoms and elevated white blood cells in the blood routine, shock is possible but requires further examination to confirm) [Human Doctor Comment]: The patient shows no signs of shock.
>
> 3. Diagnostic Basis: (1) Closed abdominal trauma: Based on the patient's history of vehicular accident and symptoms of abdominal pain, tenderness, and rebound pain. (2) Splenic injury: Color Doppler Ultrasound indicates abnormal echogenicity in the spleen, suggestive of hematoma formation and uneven echogenicity indicating possible contusion. (3) Left kidney injury: Color Doppler Ultrasound shows abnormal echogenicity in the left kidney, suggestive of hematoma formation. [Human Doctor Comment]: Not mentioned in the case. (4) Fatty liver: Diagnosed via Color Doppler Ultrasound. (5) Rib fractures: Based on the history of steering wheel loss of control and description of bilateral rib fractures. (6) Traumatic shock: Based on symptoms and elevated white blood cells, although shock is possible, further examination is needed to confirm. [Human Doctor Comment]: The patient shows no signs of shock.
>
> 4. Differential Diagnosis:
>
> 4.1. Liver rupture: Although the patient has a history of abdominal trauma and signs of peritonitis, Color Doppler Ultrasound did not indicate liver rupture, and the liver dullness boundary is intact, hence liver rupture is not considered.
>
> 4.2. Intestinal rupture: The patient has no apparent symptoms of peritonitis, and Color Doppler Ultrasound did not show any abnormality in the intestines, hence intestinal rupture is not considered.
>
> 4.3. Pancreatic injury: The patient shows no obvious symptoms such as upper abdominal pain, back pain, or elevated serum amylase, and the pancreas appeared normal on Color Doppler Ultrasound, hence pancreatic injury is not considered.
>
> 5. Final Diagnosis: Closed abdominal trauma including splenic injury, left kidney injury, [Human Doctor Comment]: Not mentioned in the case. and rib fractures.
>
> 6. Principle of Treatment: (1) Emergency management: Includes maintaining airway patency, stabilizing vital signs, and supportive treatments like fluid and blood transfusions. (2) Surgical treatment: Based on the progression of the condition, exploratory laparotomy may be required to clarify the extent of the injuries and to perform necessary repairs or resections. (3) Non-surgical treatment: For patients with mild symptoms and stable vital signs, conservative treatment including observation, fluid administration, and antibiotic use may be initiated.
>
> 7. Treatment Plan: (1) Emergency handling: Immediate oxygen administration, ECG monitoring, establishing IV access, and necessary support treatments like fluid and blood transfusions. (2) Laboratory examinations: Reevaluation of complete blood count, biochemical tests, and coagulation functions, monitoring changes in hemoglobin, platelets, and prothrombin time. (3) Imaging examinations: Abdominal CT scan if necessary [Human Doctor Comment]: Abdominal CT scan is necessary. to further clarify injury location and severity. (4) Surgical preparation: If needed, immediate preparations for exploratory laparotomy, including preparing the operating room, anesthesia, and surgical team. (5) Non-surgical treatment: For patients with stable vital signs and mild symptoms, conservative management including bed rest, vital signs monitoring, and antibiotics to prevent infection may be pursued.

Table 30: A detailed case study on ClinicalBench, diagnosed by Yi-34B-Chat.

> **Case Study (English)**
>
> (6) Intensive care: Patients with shock or other severe complications should be admitted to the ICU for close monitoring and treatment. Note that the above diagnoses and treatment plans are based on the provided case summary and examination results, and may need adjustments based on specific patient conditions and further test results. Due to the lack of pathological examination and some key laboratory test results, such as coagulation functions, the above diagnostic and treatment plans may be modified based on subsequent test results. Additionally, as the patient may have multi-organ injuries, the treatment plan should be comprehensively considered based on the severity of the injuries and the overall condition of the patient.
>
> 8. Impression: (1) Liver: Normal in shape and size, increased parenchymal echogenicity, normal diameter of the main portal vein, no abnormalities observed. (2) Gallbladder: Normal in shape and size, smooth walls, no abnormal echogenicity inside, no dilation of intrahepatic and extrahepatic bile ducts. (3) Spleen: Normal in shape and size, uneven parenchymal echogenicity, presence of a hypoechoic area, unclear boundaries, and uneven internal echogenicity. (4) Pancreas: Normal in shape and size, uniform parenchymal echogenicity, no dilation of the main pancreatic duct. (5) Kidneys: Both kidneys normal in position, shape, size, uniform distribution of parenchymal echogenicity, no separation of the collecting system, no abnormalities observed. (6) CDFI: No significant abnormalities in blood flow signals. (7) Abdominal exploration: Supine position shows no significant free fluid dark areas.

Table 31: A detailed case study on ClinicalBench, diagnosed by Yi-34B-Chat. (Cont. Table30)

