# OpenReview forum: "ClinicalLab: Aligning Agents for Multi-Departmental Clinical Diagnostics in the Real World"
_ICLR.cc/2025/Conference — ICLR 2025 Conference Withdrawn Submission_

### Official Review · Reviewer_G8Pg · 2024-10-23

**Soundness:** 4
**Presentation:** 4
**Contribution:** 3
**Rating:** 6
**Confidence:** 4

**Summary:**

Paper provides an evaluation framework called ClinicalBench, ClinicalLab, and Clinical Agent meant for evaluating LLMs on robust, multidepartment, health agents. In ClinicalBench, the authors introduce the first real-case-based, data-leakage-free, end-to-end multidepartmental benchmark for evaluating the diagnostic capabilities of LLMs across 24 departments and 150 disease types, along with four novel ClinicalMetrics to assess the practicality and effectiveness of LLM outputs, and propose ClinicalAgent, a diagnostic agent designed for modern medical settings that outperforms competitive LLMs on this benchmark.

**Strengths:**

- The paper is well motivated.
- Table 1 is helpful in giving us an understanding of the current state of clinical benchmarks.
- Easy to read.
- Very comprehensive benchmarking dataset.

**Weaknesses:**

- The # of samples appears to be on the smaller end.
- diversity in the dataset was not well described in the main paper.

**Questions:**

- How are hallucinations quantified? Is it equivalent to the incorrect responses?
- Why do you think different LLMs excel at different tasks? How can LLMs improve upon their defeciencies?
- which internLM model was used? 1.8B? 7B? 20B? This is surprising that a small LLM is beating large ones.
- I'm curious to know how diversity was measured in a Chinese based health system? Alot of concerns in healthcare revolve around biases towards underrepresented communities. In the future, some racial, age, sex biases could elicit some exciting results.
- Another interesting direction is to see if language poses any barrier in current state of the art performance? I wish in the paper, you spoke more about the subtle differences between using Chinese Based LLMs versus English.

---

### Official Review · Reviewer_KGKG · 2024-10-27

**Soundness:** 2
**Presentation:** 2
**Contribution:** 2
**Rating:** 3
**Confidence:** 4

**Summary:**

Thank you for your effort to provide comprehensive benchmarks for clinical practice. One of the strongest points is that the benchmark was designed based on several departments and diseases.  Another point is the ClinicalAgent can perform an end-to-end for real-world clinical diagnostic practices.

However, the reviewer found major concerns in the manuscript. Those concerns concern the proof that the study's claims are not convinced, the reproducibility of the proposed benchmark, and the datasets.

**Strengths:**

One of the strongest points is that the benchmark was designed based on several departments and diseases.
Another point is the ClinicalAgent can perform an end-to-end for real-world clinical diagnostic practices.

**Weaknesses:**

proof that the study's claims are not sufficient,
the reproducibility of the proposed benchmark,
and the datasets.

**Questions:**

1. It is mentioned that “LLMs still struggle to meet the strict requirements for accuracy and reliability in the medical field and face many challenges in clinical applications” ===> However, the current work does not any proof to prove that it can deal with this challenge?

Please provide specific evidence or comparative results demonstrating how the approach improves accuracy and reliability compared to existing benchmarks or LLMs in medical applications.

2. It is mentioned that “existing medical evaluation benchmarks face the risk of data leakage or contamination. And “We ensure that ClinicalBench does not have data leakage” ===> The reader can not find the detailed information to convince the claims from the author.

Please describe your methods to prevent data leakage, such as data collection procedures, preprocessing steps, or validation techniques used.

3. Existing evaluation methods are limited to multiple choice questions, which do not align with the real-world diagnostic scenarios ===> Propose Generative QA. The reviewer is not sure of the reliability of the questions; how are the composed validated sets validated by the healthcare professional practitioner? How can we follow this strategy and prepare for our in-hospital domain dataset?

4. The data processing, is not sufficient enough ===> How to deal with the preprocessing to have a complete structured data for the training and inference? For example, how to deal with numeric attributes from inside the notes?

5. Lacking of the demographic statistical analysis of the data (sex, age, etc.) Therefore, we are not certain of the generalization of the benchmark for the generated output from the agent? Especially, with the number of 1500 samples, it is actually not a large enough dataset, so we need to have more detailed how the dataset covers patient demographic statistic?

Please include a detailed demographic breakdown of their dataset, including age ranges, gender distribution, and other relevant factors. Additionally, please discuss how you ensured adequate representation across different demographic groups given the relatively small sample size.

6. We should expect to have the comparative analysis between the proposed benchmark, and the existing bechmarks from the literature? If not, it is not convinced to confirm the effectiveness of the proposed approach?

7. The most critical is the experiment setup in detailed so that the reproducibility can be made? Hyperparamters, fine-tuning approaches, etc….?

Please provide a detailed appendix or supplementary material containing complete hyperparameter settings, specific fine-tuning procedures, data preprocessing techniques, evaluation metric implementations, and code or pseudocode for key algorithms.

8. The experimental evaluation is completed through API calls and 8 NVIDIA A6000 GPUs ====> Should we have a table that compare the computational resource for the training between the proposed benchmark with different LLM models versus with existing benchmarks? Based on that, the interested readers should expect to estimated how much computational resource (training times, inference times, FLOPs, etc..) before they want to replicate the benchmarks and selected the LLM models in advance.

Please Include a comparative table showing computational resources (e.g., training times, inference times, FLOPs) for their benchmark across different LLM models and for existing benchmarks. This would help readers estimate resource requirements for replication or model selection.

**Details Of Ethics Concerns:**

In case the study is accepted for publication, how can we make sure that the datasets was obtained, prepared.
Especially, the composed set of generative QA from the study was not clearly explained or validated. Therefore, it is impossible for the interested reader want to follow the strategy to design them for performing the benchmark.

---

### Official Review · Reviewer_TLuh · 2024-11-01

**Soundness:** 2
**Presentation:** 2
**Contribution:** 2
**Rating:** 3
**Confidence:** 4

**Summary:**

This paper presents ClinicalLab, which comprises three key components: 1) ClinicalBench, an end-to-end multi-departmental clinical diagnostic evaluation benchmark designed for assessing medical agents and large language models (LLMs). ClinicalBench is based on real case data covering 24 departments and 150 diseases, ensuring no data leakage. 2) ClinicalMetrics, a set of metrics developed to evaluate the effectiveness of LLMs in clinical diagnostic tasks. 3) ClinicalAgent, an end-to-end clinical agent aligned with real-world clinical diagnostic practices.

**Strengths:**

1. This paper addresses a significant and timely topic by focusing on the performance evaluation of large language models (LLMs) in clinical diagnosis within the medical field. The relevance of this focus is crucial, given the increasing adoption of AI technologies in healthcare.
2. The authors have constructed a valuable real-world dataset that encompasses 24 departments and 150 diseases. This comprehensive dataset is a substantial asset for advancing the development of intelligent solutions in the medical domain, providing a solid foundation for future research and application in clinical diagnostics.

**Weaknesses:**

1. This paper addresses important challenges related to medical clinical datasets and evaluation metrics for auxiliary diagnostic tasks, which is a significant contribution to the field. However, it appears that the proposed solutions, particularly ClinicalMetrics, do not introduce substantial technical innovation or new perspectives in evaluation. The lack of novel insights raises questions about the overall advancement of the metrics presented. Further exploration of innovative approaches would enhance the contribution of this work to the literature.
2. The creation of a comprehensive real-world dataset encompassing 24 departments and 150 diseases is a notable contribution of this work. However, I would like to understand whether there are plans to release this dataset to the public in the future. If public access is not currently feasible, could the authors provide specific reasons for this limitation, such as privacy concerns or data governance issues?

**Questions:**

1. Given the specific focus on dataset development and evaluation metrics, could this paper be more appropriately situated for publication in specialized journals dedicated to medical informatics or clinical data analysis?

---

### Official Review · Reviewer_6dhy · 2024-11-03

**Soundness:** 3
**Presentation:** 3
**Contribution:** 3
**Rating:** 6
**Confidence:** 4

**Summary:**

This paper introduces ClinicalLab, a comprehensive suite for aligning and evaluating clinical diagnostic agents. The main contributions include: a) ClinicalBench: A new end-to-end, multi-departmental clinical diagnostic evaluation benchmark based on real cases, covering 24 departments and 150 diseases. b) ClinicalMetrics: Four novel metrics for evaluating the effectiveness of language models in clinical diagnostic tasks. c) ClinicalAgent: An end-to-end clinical agent designed to align with real-world clinical diagnostic practices. The authors evaluate 17 general and medical-domain language models on ClinicalBench and propose ClinicalAgent as an improved approach for clinical diagnostics.

**Strengths:**

This paper addresses important limitations in existing medical AI benchmarks, which is a unique contribution. The dataset includes comprehensive coverage of medical departments and diseases. In addition, they propose novel metrics for evaluating clinical diagnostic capabilities. From the dataset and benchmark perspective, they leverage real-world clinical data in a private dataset, which is not present in previous datasets, as shown in Table 1. They also thoroughly evaluate multiple language models and propose an aligned clinical agent system.
- The work addresses critical gaps in evaluating AI models for clinical applications, potentially improving patient care and medical decision-making.
- The benchmark and agent align closely with real-world clinical practices, making the research highly relevant to both AI and medical communities.
- The research demonstrates rigorous methodology in data collection, metric development, and model evaluation.
- The authors show strong attention to patient privacy and ethical use of medical data with sufficient materials support in the supplementary materials submission.

**Weaknesses:**

-  To me ClinicalAgent works as a medical domain-specific LLM without planning or tool-use, instead of LLM-based agents (e.g., AgentMD, EHRAgent, etc.).
- If it works as a multi-agent system, then it might need further comparisons with other medical agents like MedAgent, MDAgents, etc.
- The proposed ClinicalMetrics, such as CDR, Acceptability, and DWR, are essentially combinations of existing metrics and should not be considered novel. DIFR is better characterized as a metric assessing instruction-following capabilities. Furthermore, BLEU, ROUGE, and BERTScore are not optimal for evaluating medical generative question-answering tasks, as medical cases often contain numerous abbreviations that can interfere with these metrics’ effectiveness.

**Questions:**

See above.

---

### Official Review · Reviewer_rBWR · 2024-11-04

**Soundness:** 2
**Presentation:** 2
**Contribution:** 3
**Rating:** 3
**Confidence:** 4

**Summary:**

The paper introduces ClinicalLab, a comprehensive suite for aligning clinical diagnosis agents, which includes ClinicalBench, ClinicalMetrics, and ClinicalAgent. ClinicalBench is aimed at providing an end-to-end evaluation benchmark, while ClinicalMetrics comprises four novel metrics designed to assess the effectiveness of medical agents and LLMs. Additionally, the authors present ClinicalAgent, a new clinical diagnostic agent developed in response to the absence of a single expert LLM.

**Strengths:**

- The new large-scale evaluation benchmark for LLMs in the medical domain could be highly beneficial to the research community.
- A variety of LLMs were evaluated in this paper, allowing researchers to observe trends and understand which models are operational in various tasks.

**Weaknesses:**

- Although the ClinicalBench benchmark is described as an end-to-end clinical diagnostic evaluation benchmark for assessing medical agents and LLMs (as presented in the abstract, L23-L24), the paper does not demonstrate an end-to-end clinical diagnostic evaluation for each LLM, nor is it clear how this evaluation is conducted. Specifically, according to Sections 3 and 4, all LLMs are evaluated based on their performance in eight separate tasks (using ground-truth input), rather than in an end-to-end manner. In other words, this benchmark can be viewed as a collection of QA benchmarks (with defined inputs and outputs) rather than as a benchmark for an end-to-end agent capable of sequential decision-making.

- Some metrics are redundant or unclear. Instead of the DWR metric, simply using micro accuracy (across departments) may be a better metric to show absolute values and map rankings. Additionally, the DIFR metrics are more related to instruction-following ability rather than clinical performance metrics. Furthermore, during the ClinicalBench evaluation, only one department is relevant, so it is unclear why the DIFR metric holds importance in this benchmarking context.

- The evaluation mechanisms for ClinicalAgent (Section 5) and the LLMs (Sections 3 and 4) differ, meaning that only the end-to-end evaluation results are conducted with ClinicalAgent, not the 17 LLMs. Specifically, the LLMs in Table 4 are evaluated based on individual abilities and do not make sequential predictions (the GPT-4o score is the same in both Table 2 and Table 4).

**Questions:**

- Could you provide the prompt formulation for each of the 8 tasks? I cannot find how the prompts for evaluation are formulated for each task in the manuscript or Appendix.
- For Sections 3.4.1, 3.4.2, and 3.4.3, is the input information based on ground-truth data? Additionally, do these three major function groups operate independently of one another?
- The equation for DWR might be incorrect. It is intended to measure the results of each LLM, but the equation aggregates the scores of all LLMs.
- What are the values for K' in the DIFR metric used in this study? Also, in the CDR metric, what are the disease diagnoses mentioned in L360? Additionally, how is it determined whether the prediction for disease diagnosis is correct? Is it simply a strict match between the ground-truth and the generated text (followed by your postprocessing technique for medical synonyms)?
- For the automatic evaluation of ClinicalAgent, could you provide the full experimental results similar to Table 2, including each of the 8 tasks?
- For the human or GPT-4o evaluation, do you assess the intermediate results produced by the agent, such as laboratory examinations or a set of tests? This is important in relation to the end-to-end evaluation of clinical agents.

---

### Note · Authors · 2024-12-16

I have read and agree with the venue's withdrawal policy on behalf of myself and my co-authors.